# Targeted CRISPR-Cas9 screening identifies core transcription factors controlling murine haemato-endothelial fate commitment

Michael Teske [1,2], Tobias Wertheimer[1,10], Stefan Butz[2], Pascale Zwicky[1], Izaskun Mallona [3], Svenja L. Nopper[1], Christian Münz[1], Ulrich Elling [4,11], Christophe Lancrin [5], Burkhard Becher [1], Ana Rita Grosso [6,7], Tuncay Baubec [2,8] & Nina Schmolka [1,2,9] ✉

During development, blood generation begins in the yolk sac with the differentiation of haemato-endothelial mesoderm forming haematopoietic progenitors. This study aims to identify the crucial molecular regulators of haemato-endothelial mesoderm formation and to extend our knowledge of the process in an unbiased way. We employ a murine embryonic stem cell model that recapitulates embryonic blood development, and perform targeted CRISPR-Cas9 knock out screens focusing on transcription factors and chromatin regulators. We identify the transcription factors ETV2, LDB1, SMAD1, SIX4 and ZBTB7b as regulators of haemato-endothelial mesoderm commitment. Embryonic stem cells lacking these regulators give rise to mesodermal subsets with a defined lineage differentiation bias, while transcriptome analysis of these cells uncovers the precise impact of each factor on gene expression in the developing mesoderm. Our study reveals molecular pathways governing mesodermal development crucial to allow endothelial and haematopoietic lineage specification and paves the way for future advances in haematopoietic stem cell applications.

Lineage commitment is a complex and multi-factorial process that is achieved by the stepwise acquisition of cell fate changes, which begin in the pluripotent cells of the embryo. Within these cells, emergence of identity results from a combination of encoded genetic and structural chromatin-based information that ultimately determines individual gene expression programmes: while genetic information encodes transcription factors (TFs) that drive progenitor fate and differentiation, chromatin-based information shapes the accessibility of DNA to those TFs, providing a heritable chromatin landscape that can direct or reinforce lineage-specific transcriptional programmes.

Understanding the process by which blood cells are generated is particularly important. The comprehensive identification of factors that specify haematopoietic fate during mesodermal development not only provides valuable insights into lineage specification and

[1]Institute of Experimental Immunology, University of Zurich, Zurich, Switzerland. [2]Department of Molecular Mechanisms of Disease, University of Zurich, Zurich, Switzerland. [3]Department of Molecular Life Sciences, University of Zurich, Zurich, Switzerland. [4]Institute of Molecular Biotechnology of the Austrian Academy of Science (IMBA), Vienna BioCenter (VBC), Vienna, Austria. [5]European Molecular Biology Laboratory, EMBL Rome - Epigenetics and Neurobiology Unit, Monterotondo, Italy. [6]UCIBIO-Applied Molecular Biosciences Unit, Department of Life Sciences, NOVA School of Science and Technology, Universidade NOVA de Lisboa, Caparica, Portugal. [7]Associate Laboratory i4HB—Institute for Health and Bioeconomy, NOVA School of Science and Technology, Universidade NOVA de Lisboa, Caparica, Portugal. [8]Genome Biology and Epigenetics, Institute of Biodynamics and Biocomplexity, Department of Biology, Utrecht University, Utrecht, The Netherlands. [9]Universidade Católica Portuguesa, Católica Medical School, Católica Biomedical Research Centre, Lisbon, Portugal. [10]Present address: Department for Internal Medicine I, University Medical Center Freiburg, Freiburg, Germany. [11]Present address: Viverita Discovery, Vienna, Austria. ✉e-mail: nina.schmolka@uzh.ch

reprogramming but is essential for the generation of haematopoietic progenitors from pluripotent stem cells for regenerative medicine. Accordingly, the regulation of the process of early haematopoietic specification in the embryo has been extensively studied in recent years. In mice, blood development initiates at around embryonic day 7 in the blood island of the yolk sac[1,2]. Here, blood cells develop from a mesodermal precursor with both haematopoietic and endothelial developmental potential: the haemato-endothelial mesoderm (HEM) or haemangioblast[3–6]. The HEM, in turn, initiates both primitive and definitive haematopoiesis by giving rise to primitive haematopoietic progenitors (pHPCs) and haemogenic endothelial cells (HECs), respectively[7,8]. pHPCs further differentiate to primitive blood cells, including primitive erythrocytes and primitive macrophages[1,2,9]. HECs initiate definitive blood development by undergoing an endothelial-to-haematopoietic transition (EHT), where, in a second wave of blood formation, mainly erythro-myeloid progenitors (EMPs) are formed[10]. Upstream HEM develops from an early common mesoderm progenitor, called primitive mesoderm (priMes), that has a broad developmental potential for both the cardiac lineage and the haemato-endothelial lineage[11,12]. Therefore, the transition from priMes towards HEM is the first transition that locks a cell into a haemato-endothelial fate. Several transcription factors that affect mesodermal commitment towards a haemato-endothelial fate have been identified: Brachyury (T) and Eomesodermin (EOMES) effect changes to the chromatin that are crucial for the initial stages of pluripotent cell commitment towards a mesodermal fate[13]; while downstream, ETV2 is considered the master regulator of the transition of the multipotent priMes towards HEM. While loss of function of ETV2 results in a complete differentiation block of both lineages and early lethality[14,15], ectopic expression of ETV2 enables fibroblasts to reprogram towards an endothelial or haematopoietic fate[15–17]. A recent study employing hypomorph ETV2 mutants highlighted a differential dose-dependency for lineage commitment, with haematopoietic lineage commitment being more sensitive to reduced ETV2 dosage than the endothelial lineage[18]. We also know which factors are further needed for full blood commitment after haemato-endothelial differentiation occurred, including TAL1, RUNX1, and GATA2[19]. The major open question in the field today is how these—and possibly other yet-unidentified—regulators cooperate to allow the initial transition towards the blood lineage by the differentiation of priMes to the HEM.

Here, we answered these questions using a mouse embryonic stem cell (ESC) differentiation model that accurately recapitulates mesoderm development and embryonic haematopoiesis in the yolk sac[7,20,21], combined with large-scale CRISPR-Cas9 KO screens focusing on transcription factors and chromatin regulators. We confirmed ETV2 as a master regulator driving primitive towards haemato-endothelial mesoderm commitment and identified several additional transcriptional regulators playing key roles in lineage specification and its repression, including LDB1, SMAD1, SIX4, and ZBTB7b, respectively. Loss-of-function studies of the newly identified core TFs revealed profound perturbations to normal haematopoiesis, further confirming the significance of our findings. Collectively, by employing large-scale CRISPR screening, we identify molecular regulators impacting on haematopoietic-endothelial lineage commitment.

## Results

### In depth characterisation of the developmental route to HEM

In vitro murine ESC differentiation recapitulates the stages of mesoderm differentiation to HEM—the precursor of blood and endothelial lineages—commitment in vivo[21–23] (Supplementary Fig. 1a). Distinct mesodermal entities form during embryoid body (EB) culture and two surface markers, FLK1 (vascular endothelial growth factor receptor 2 (VEGFR2)) and PDGFRα (platelet-derived growth factor receptor alpha), distinguish distinct differentiation stages. FLK1 together with PDGFRα marks primitive mesoderm, a multipotent mesodermal

population with a broader developmental potential (DP priMes), whereas haemato-endothelial mesoderm is single-positive for FLK1 (F_SP HEM). We first performed an in-depth phenotypic and molecular characterisation of the developing mesodermal populations to test and confirm the validity of this model. Whereas uncommitted cells do not express FLK1 or PDGFRa (DN), mesodermal commitment started at day 4 with the detection of P_SP cells and DP priMes; by day 5 all three types of mesoderm were present (P_SP, DP priMes, and F_SP HEM) (Fig. 1a, b). To confirm the developmental trajectory, we conducted gene expression analysis by single-cell RNA sequencing employing SORT-seq, which we then used to construct the developmental trajectory from DN → P_SP → DP priMes → F_SP HEM populations (Fig. 1c and Supplementary Fig. 1b, c). This analysis aligned with previous reports employing ESC differentiation models[23,24] and with the kinetics of FLK1 and PDGFRα expression in our EB culture. To further validate the use of FLK1 and PDGFRα expression patterns to define mesodermal populations, we performed bulk RNA-seq on DN, P_SP, DP priMes, and F_SP HEM populations. Principal component analysis separated the four populations (Supplementary Fig. 1d) and allowed us to confirm the expression of relevant marker genes for distinct developmental stages: pluripotency genes (e.g. *Nanog, Pou5f1, Fgf5*) were expressed in DN cells, early mesodermal markers (e.g. *Mesp1, Mesp2, Mixl1*) in P_SP and DP priMes cells, and haematopoietic and endothelial genes (e.g. *Lmo2, Tal1, Sox7*) in F_SP HEM cells (Supplementary Fig. 1e).

As F_SP HEM is further differentiated towards the haemato-endothelial fate compared to the other mesodermal populations, F_SP HEM should display an increased developmental potential to form blood and endothelial lineages. To examine this, we isolated DN, P_SP, DP priMes, and F_SP HEM cells by FACS and placed them into haemangioblast (HB) culture to support their differentiation to haematopoietic and endothelial progenitors as well as vascular smooth muscle cells (VSM)[6,25]. After 2 days, we used CD41 and VE-cadherin (VE-cad) expression to distinguish haematopoietic (CD41+), endothelial (VE-cad+), pre-haematopoietic stem and progenitor cells (pre-HSPC, CD41+VE-cad+), and VSM cells (CD41-VE-cad-)[26] by flow cytometry. As there is no specific surface marker for VSM cells suitable for flow cytometry, we will refer to CD41-VE-cad- cells as VSM-enriched, acknowledging the inferred nature of this identity. The HB culture experiments confirmed that DP priMes indeed had less capacity to form haematopoietic and endothelial lineages compared to F_SP HEM, which generate a higher frequency of CD41+ and VE-cad+ cells and fewer VSM-enriched cells (Fig. 1d, e). Accordingly, and as expected, DN and P_SP had a further reduced capacity to form haematopoietic and endothelial lineages compared to DP priMes/ F_SP HEM (Fig. 1d, e). To examine the lineage potential of the two different mesodermal populations at higher resolution, we next established a high-dimensional flow cytometry panel using antibodies recognising 23 surface markers (Supplementary Table 1). We performed unsupervised clustering to define nine major clusters based on their expression patterns. F_SP HEM was more proficient at forming haematopoietic and endothelial lineages, including primitive (primitive erythrocyte progenitors (pEryP, CD41+/- CD71+c-kit-) and definitive blood subsets comprised of definitive haematopoietic progenitors (dHPC, CD41+/-c-kit+CD71+CD43+), pre-haematopoietic stem and progenitor cells (pre-HSPC, CD41+VE-cadTie2+CD34+c-kit+) and macrophage progenitors (MacP, CD41+/-c-kit+/-CD11b/c+CD44+CD45+Cx3Cr1+) and endothelial type 1 (Tie2+FLK1+CD31+VE-cad+c-kit−) and type 2 (Tie2+FLK1+CD31+VE-cad+c-kit+) clusters, whereas DP priMes had a higher potential to form VSM-enriched cells (CD41−VE-cad−CD71−Epcam−) (Fig. 1f, g and Supplementary Fig. 2). To further assess the functional differences of DP priMes and F_SP HEM, we established a long-term haematopoietic assay to promote the maturation of haematopoietic progenitors. We compared (i) the blood-forming potential and (ii) the ability to generate mature blood lineages of these two mesodermal populations. Defined numbers of

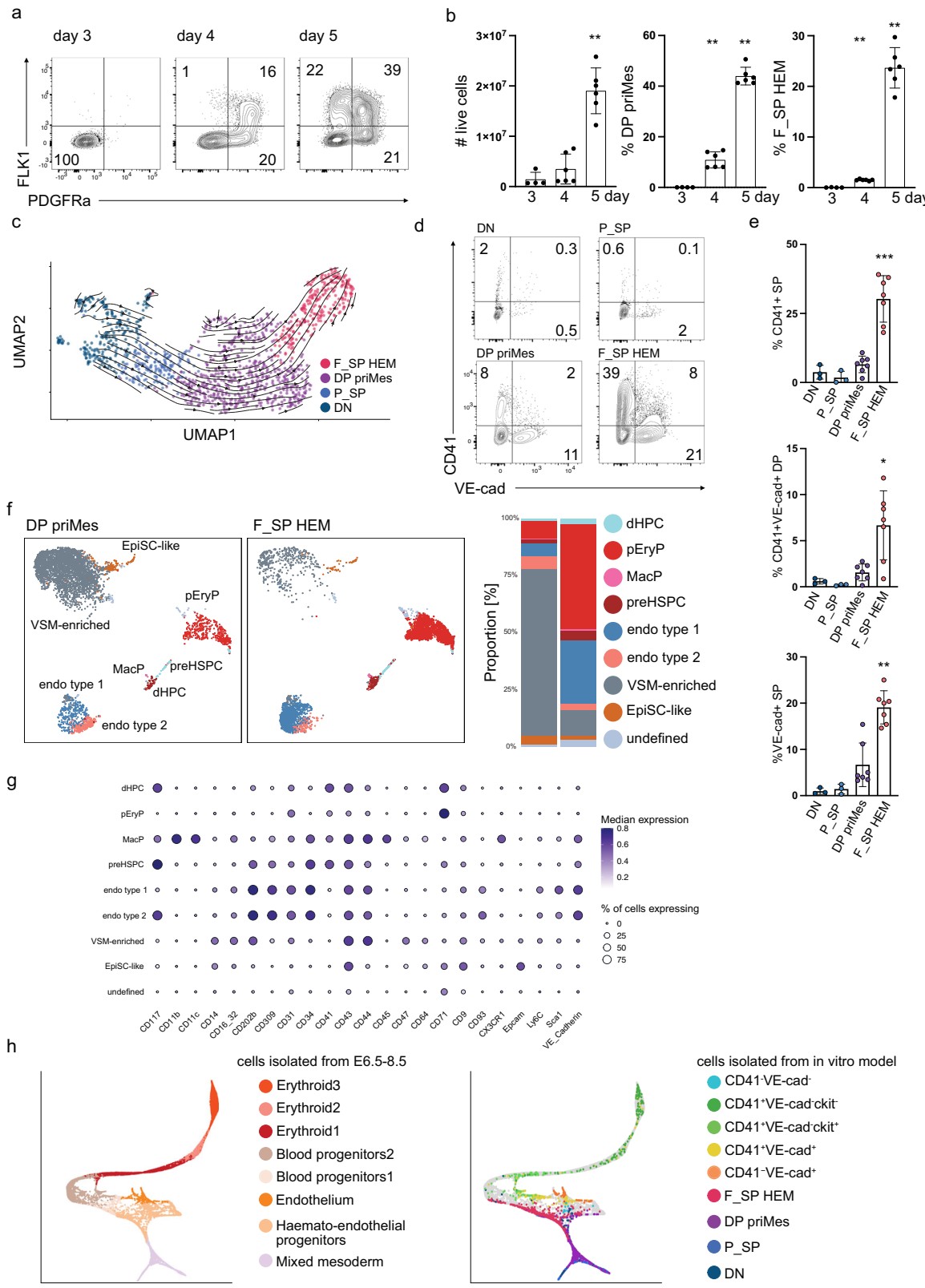

FACS-sorted DP priMes and F_SP HEM (18k, 10k, or 5k) isolated from EB cultures were plated and analysed after 9.5 days. Whereas F_SP HEM generated haematopoietic cells under all tested conditions, DP priMes showed reduced blood-forming potential, particularly at lower cell inputs: For 5k cells, 100% of F_SP HEM formed colonies, but only 8% of DP priMes (Supplementary Fig. 3a, b). Additionally, for blood lineages analysis, we scored 8 randomly selected wells for each condition and

analysed the number of live cells and the frequency of leukocytes (CD45+) and erythroid (CD71+) cells, the two main blood lineages forming in the assay (Supplementary Fig. 3c–f). F_SP HEM consistently showed higher number of live cells and increased frequencies of CD45+ and CD71+ cells in all analysed conditions compared to DP priMes (Supplementary Fig. 3c-f). Collectively, these results demonstrate that mesodermal populations, defined by FLK1/PDGFRα

**Fig. 1 | Characterisation of murine ESC-derived differentiation model towards the haemato-endothelial cell fate. a** Representative flow cytometry analysis of FLK1 and PDGFRα surface expression in EB cultures between days 3 and 5 of culture. **b** (left) Number of live cells (day4 vs. day5 *p* = 0.0022) and (middle) percentage of FLK1+/ PDGFRα + (DP) cells (day3 vs. day4 *p* = 0.0095, day4 vs. day 5 *p* = 0.0022) and (right) FLK1+/ PDGFRα - (F_SP) cells in EB cultures over time (day3 vs. day 4 *p* = 0.0095, day4 vs. day5 *p* = 0.0022). **c** UMAP of transcriptomic profiles of 1221 cells isolated by FACS from EB cultures at day 5 and analysed by scRNA-seq. Cells are coloured by EB population, as defined by surface marker expression. Arrows represent RNA velocity vectors, indicating predicted developmental trajectories between populations. **d** Representative flow cytometry analysis of CD41 and VE-cadherin (VE-cad) surface expression and (**e**) frequency of CD41 + SP, CD41+/VE-cad+ DP and VE-cad+ SP cells of HB cultures started from mesodermal populations. **f** (left) UMAP with overlaid FlowSOM clustering of 50,000 cells and (right) bar chart depicting proportions of cellular subsets identified by FlowSOM

clustering of cells and **g** Expression matrix of markers used in high-dimensional flow cytometry analysis from HB culture started from either DP priMes or F_SP HEM. **h** (left) Force-directed graph of cells isolated from E6.5 to E8.5 embryos associated with the blood/endothelial lineage (adapted from[27]). Plot highlights various cell types that are generated along the haemato-endothelial lineage trajectory as mesoderm differentiates into endothelial and haematopoietic cells. (right) Mapping of in vitro-generated cells isolated from EB and HB cultures. Each data point (in **b** and **d**) represents an individual experiment (**b**) *n* = 4 biological replicates, (**d**) *n* = 3 biological replicates. Error bars represent mean ± SD. *P* values were calculated using an unpaired, two-tailed *t*-test (Mann-Whitney). SP single-positive, DP double-positive, EpiSC epiblast stem cell, pEryP primitive erythrocyte progenitors, preHSPC pre-haematopoietic stem progenitor cells, dHPC definitive haematopoietic progenitor cells, MacP macrophage progenitors, VSM vascular smooth muscle, endo endothelia. Source data for (**b**, **e**, **f**) are provided as a Source data file.

expression, differ significantly in their blood-forming capacity. F_SP HEM is functionally distinct from DP priMes with superior potential to form mature haematopoietic cells due to increased potential to form haematopoietic progenitors. This highlights the importance of FLK1/ PDGFRα expression in marking functionally distinct mesodermal subpopulations during early haematopoietic development.

Lastly, to further underline the relevance of our ESC model, we performed a scRNA-seq characterisation of both stages of our model employing SORT-seq. By relying on the lineage defining markers FLK1, CD41, VE-cad, and c-kit, we isolated single cells corresponding to 7 distinct populations from the EB and HB stage (Supplementary Fig. 4a). Our analysis confirmed the generation of distinct lineages characterised by appropriate signature gene expression patterns, including pHPCs (primarily primitive erythrocyte progenitors), dHPCs, VSM-enriched cells, and endothelial lineage (Supplementary Fig. 4b–d). Additionally, we integrated the transcriptional profiles of our in vitro-generated cells with a mouse gastrulation atlas[27], which includes cells isolated from E6.5–E8.5 embryos. A high proportion (39.8%) of the in vitro-generated cells could be directly projected onto the in vivo haemato-endothelial trajectory graph, evidencing the similarities between in vitro- and in vivo- generated cells (Fig. 1h). Taken together, our functional analysis, single-cell and bulk RNA-seq profiles confirm FLK1 and PDGFRα as appropriate markers to distinguish mesodermal populations and attest that the model faithfully recapitulates early mesodermal commitment and subsequent blood differentiation.

## CRISPR screens identify the core transcription factors of HEM commitment

Having confirmed the developmental route giving rise to F_SP HEM, we performed loss-of-function CRISPR-Cas9 screening to identify factors regulating the transition from DP priMes towards F_SP HEM. We first established the CRISPR screen workflow using a library against transcription factors and chromatin regulators, which targeted 4077 genes using sets of five individual guide RNAs (EpiTF library) (Fig. 2a). We transduced the sgRNA library containing a GFP reporter into an ESC line constitutively expressing Cas9[28] to ensure optimal KO efficiency. Two days after transduction, sgRNA-expressing (GFP+) ESCs were sorted and used to initiate EB differentiation. We further confirmed the single guide representation at the start of the culture (Supplementary Fig. 5a). On day 5, we collected the cells and isolated the four mesodermal populations using FLK1 and PDGFRα expression (Fig. 2a). We then recovered and sequenced sgRNAs from the mesodermal populations and ranked them using MAGeCK. To discover genes that regulate the transition from DP priMes towards F_SP HEM, we compared sgRNA presentation in these two mesodermal populations. We generated an integrated ranked candidate gene list by calculating the rank mean of the three individual screens conducted in this way (Fig. 2b, c). By pinpointing "hits" where the targeting sgRNA abundance was either depleted or enriched in the F_SP HEM versus DP priMes population

with high statistical ranking, we identified genes that likely act as potential drivers or repressors for this lineage transition, respectively (Fig. 2b, c). We additionally scored the top 200 depleted genes of all mesodermal transitions and identified common overlaps in the three independent CRISPR screens (Supplementary Data 1). Focusing on the transition of DP priMes towards F_SP HEM we identified 10 common overlaps for the top 200 depleted genes (Fig. 2d): *Smad1, Etv2, Ep3O0, Rps27a, Mis18a, Ldb1, Rpa1, Med18, Six4, Hira*. For the top 200 enriched genes, we identified four overlaps (Fig. 2e): *Cop1, Zbtb7b, Zic3*, and *Npm1*. Interestingly, all overlapping factors displayed a high screen performance and were among the top 15 depleted/enriched genes, respectively (Fig. 2c). As we also detected sgRNA in the populations T0 (ESCs collected at the beginning of the EB culture), DN, and P_SP, we could ask whether the identified "hit genes" had differences in guide abundance (cpm reads) specifically in the transition from DP towards F_SP mesoderm (Fig. 2f and Supplementary Fig. 5b). This was the case for the 14 overlaps that we detected in the individual screens. Importantly, ETV2, the best-known TF regulating DP priMes towards F_SP HEM transition, was among the top hits of the depleted genes (rank 2) serving as a positive control for our screening set-up.

## Validation of Hit Genes that promote or repress haemato-endothelial mesoderm commitment

To prioritise hit genes for validation, we applied three selection criteria: genes had to be in the top 15 factors of most depleted or most enriched; genes had to be common to the top 200 hits of the three independent screens; and genes had to have a *p* value < 0.001. This resulted in a potential candidate list of 14 genes (10 depleted genes and four enriched genes) (Supplementary Fig. 6a). To assess whether those genes regulated F_SP HEM commitment, we cloned independent KO ESC lines and performed EB differentiation experiments to score their ability to form F_SP HEM. Importantly, we generated at least three independent KO lines for each candidate to ensure that their phenotype was indeed driven by the gene KO and compared them to controls (Ctrls) transfected with non-targeting guides. For three genes, Mis18a (kinetochore protein), Rps27a (ribosomal protein), and Rpa1 (cell cycle protein), we were not able to generate viable KO ESC lines, pointing to an essential role of those genes, and we had to exclude them from further analysis. Of the seven depleted candidates, four resulted in a significant decrease in the frequency of F_SP HEM: *Smad1* (rank 1), *Etv2* (rank 2), *Ldb1* (rank 6), and *Six4* (rank 9) (Fig. 3a, b). The KO lines exhibited an accumulation of cells at earlier developmental transitions: *Etv2* KO, *Smad1* KO, and *Ldb1* KO had a significant increased frequency of DP priMes and *Six4* KO of DN cells (Supplementary Fig. 6b–d).

To understand those genes' impact on cellular differentiation, we further analysed the frequency of total FLK1+ cells (including both DP priMes and F_SP HEM) and calculated the ratio of F_SP HEM/ total FLK+ cells in cultures derived from the KOs (Fig. 3c and Supplementary Fig. 6e). Importantly, cultures generated from KOs of our four

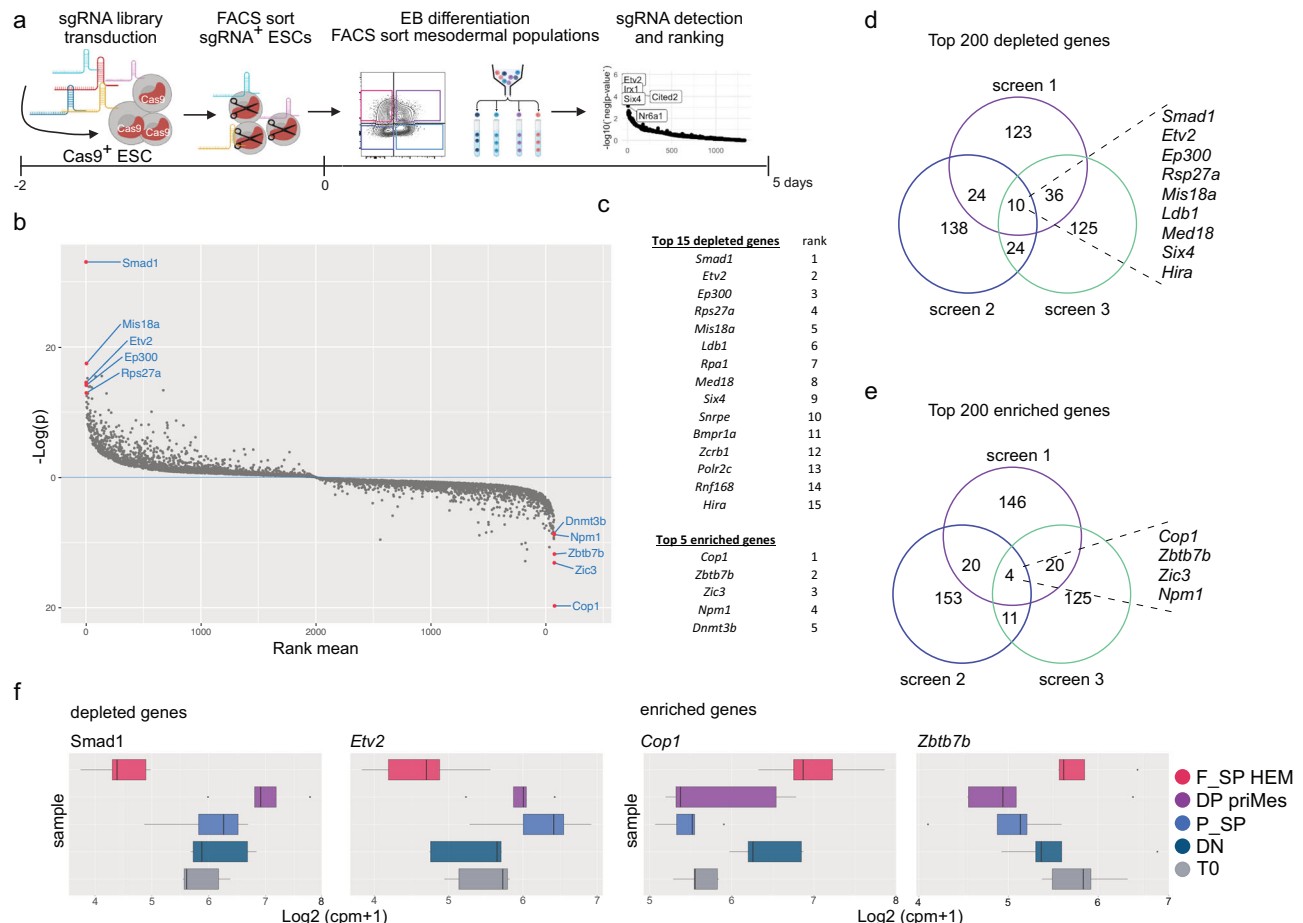

**Fig. 2 | Pooled targeted CRISPR-Cas9 screens identify drivers and repressors of HEM differentiation. a** Schematic of pooled targeted CRISPR-Cas9 screens in EB cultures. **b** Integrated rank and p-values from MAGeCK gene-level analysis of three independent CRISPR-Cas9 screens for the transition of DP priMes towards F_SP HEM. p-values were calculated by MAGeCK (permutation test) and combined by fisher's method to integrated *p*-values. **c** List of top candidate genes of three independent CRISPR-Cas9 screens for the transition of DP priMes towards F_SP HEM. **d** Overlaps of top 200 depleted genes and **e** overlaps of top 200 enriched genes in three individual CRISPR-Cas9 screens for the transition of DP priMes

towards F_SP HEM. **f** Representative count per million (CPM) values of sgRNA abundance of one CRISPR-Cas9 screen at T0 and mesodermal populations of top two depleted and enriched genes: Smad1, Etv2, Cop1, and Zbtb7b. Box plot represents upper and lower quartiles. Whiskers extend to the smallest and largest observed values within Q1−1.5 × IQR and Q3 + 1.5 × IQR; values beyond these limits are plotted as outliers. Central line is the median. *n* = 3 biological replicates. Source data for (**f**) are provided as a Source data file. **a** created in BioRender. Schmolka, N. (2025) https://BioRender.com/45x7fir.

validated candidates generated similar frequencies of total FLK1+ cells compared to the control that resulted in a significant decrease in the ratio of F_SP HEM/ total FLK+ cells (Fig. 3b, c). This points to a specific role of these genes in regulating the transition of DP priMes towards F_SP HEM and not to a general defect of FLK1+ mesodermal commitment. In addition, no significant differences in the total number of live cells were detected between KO cultures or compared to controls, ruling out an effect on cell proliferation/viability (Fig. 3d). Surprisingly, the gene KOs for *Ldb1*, *Smad1*, and *Six4* had an equally strong effect on arresting DP to further differentiate towards F_SP, as the known TF ETV2. We subsequently term these validated TFs F_SP HEM-driving TFs. Additionally, to understand if the identified TF factors lead to a delayed maturation of F_SP HEM differentiation, we extended the EB culturing time (Supplementary Fig. 6f–h). We could not detect any increase of F_SP HEM on EB day 6 in the KO cultures compared to day 5 and conclude that the identified F_SP HEM-driving TFs lead to disruption or block in differentiation, not a shift in timing (Supplementary Fig. 6f–h).

We next considered the enriched genes, analysing four hits and validating one: *Zbtb7b* (rank2). EB cultures of *Zbtb7b* KO cells generated F_SP HEM at a significantly higher frequency than controls, leading to an increased ratio of F_SP HEM/ total FLK+ cells (Fig. 3e–g).

Zbtb7b KO cells showed decreased frequencies of earlier transitions, including DP priMes and P_SP (Supplementary Fig. 6i–k). Importantly, Zbtb7b KO cultures contained a comparable number of live counts and total FLK1+ cells as did controls (Fig. 3h and Supplementary Fig. 6l). These results suggest that Zbtb7b KO cells have an enhanced potential to commit to F_SP HEM at the expense of DP priMes/ P_SP, without affecting cell proliferation: Zbtb7b therefore functions as a repressor of F_SP HEM differentiation. Uniquely, EB cultures of *Npm1* KO cells resulted in significantly fewer live cells, but F_SP HEM and DP priMes still formed, pointing to a general function of this gene in cell proliferation (Fig. 3e, h and Supplementary Fig. 6i). Due to the low number of surviving cells, significantly fewer F_SP HEM and DP priMes were generated, but a similar ratio of F_SP HEM/total FLK1+ cells were achieved compared to the Ctrl (Fig. 3e, g, h). These results agree with previous reports, which showed that Npm1 suppression inhibits ESC proliferation but does not have a general effect on cell differentiation of neural stem cells[29,30]. As we aimed to identify factors that act specifically on the differentiation of F_SP HEM, we excluded NPM1 from further analysis.

Taking advantage of our scRNA-seq dataset, we then analysed the gene expression of the validated candidates along the mesodermal differentiation trajectory. In agreement with previous reports[31], *Etv2*

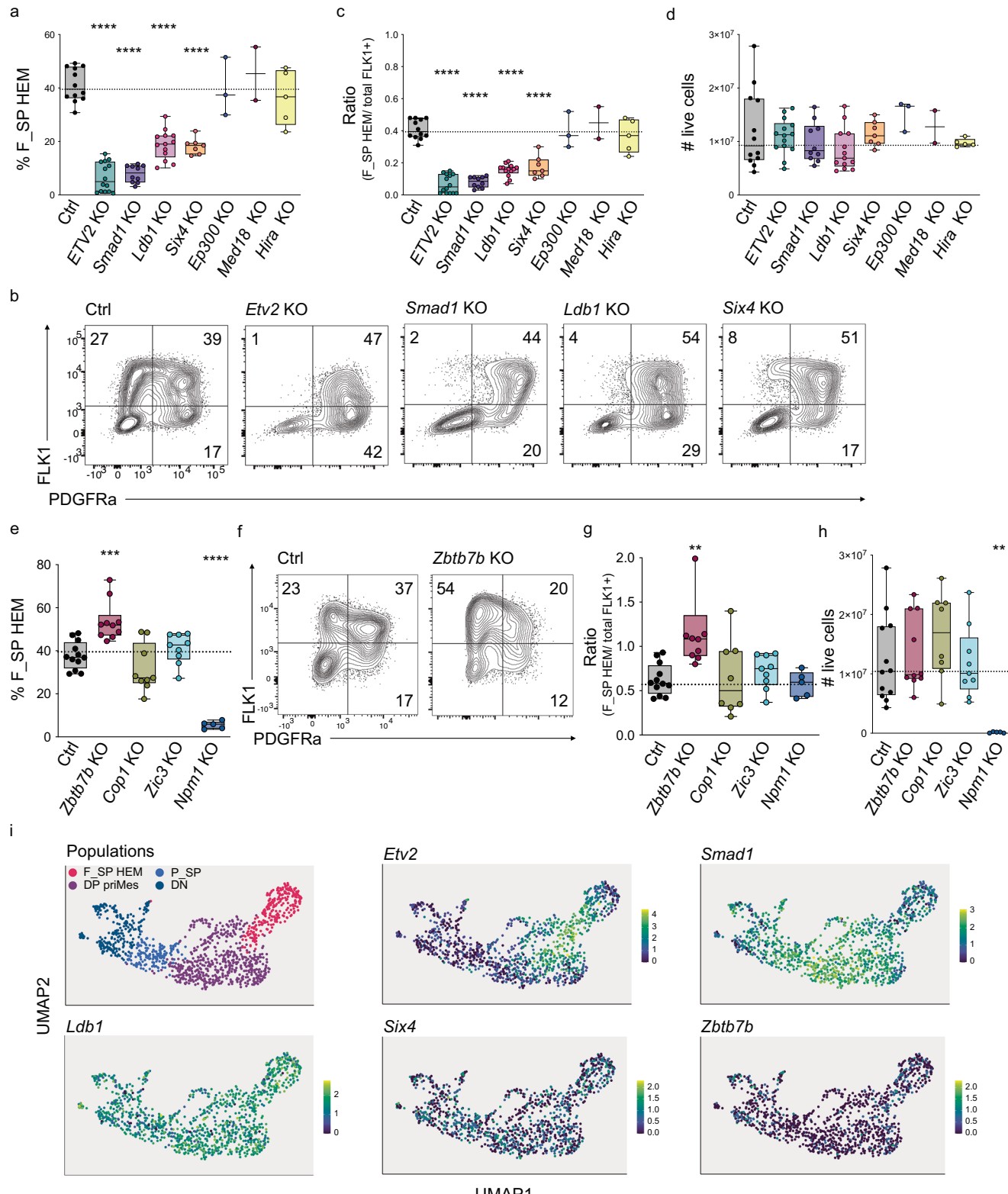

**Fig. 3 | Validation of *Ldb1*, *Six4*, *Smad1*, and *Ztbt7b* as regulators of HEM differentiation. a**, **e** percentage of F_SP HEM in EB cultures from indicated KO cell cultures (for Ctrl vs. *ETV2* KO, Ctrl vs. *Smad1* KO, Ctrl vs. *Ldb1* KO, Ctrl vs. *Six4* KO, all $p < 0.0001$; for Ctrl vs. *Zbtb7b* KO $p = 0.007$, for Ctrl vs. *Npm1* KO $p < 0.0001$). **b**, **f** Representative flow cytometry analysis of FLK1 and PDGFRα surface expression in EB cultures from indicated KO cells. **c**, **g** Ratio of F_SP HEM/ total FLK+ in EB cultures from indicated KO cells. (for Ctrl vs. *ETV2* KO, Ctrl vs. *Smad1* KO, Ctrl vs. *Ldb1* KO, Ctrl vs. *Six4* KO all $p < 0.0001$; for Ctrl vs. *Zbtb7b* KO $p = 0.0012$).

**d**, **h** Number of live cells in EB cultures from indicated KO cells (for Ctrl vs. *Npm1* KO $p = 0.0011$). **i** Expression of validated candidate genes in mesodermal populations analysed by scRNA-seq. Each data point (in **a**, **c**, **d**, **e**, **g**, **h**) represents an individually generated culture. For each cell line, three independent KO clones were analysed, $n = 4$ biological replicates. *P* values were calculated using a one-way ANOVA multiple comparisons analysis. Box plot whiskers represent the minimum and maximum value. Bounds mark the 25th and 75th percentiles. Central line is the median. Source data for (**a**, **c**–**e**, **g**, **h**) are provided as a Source data file.

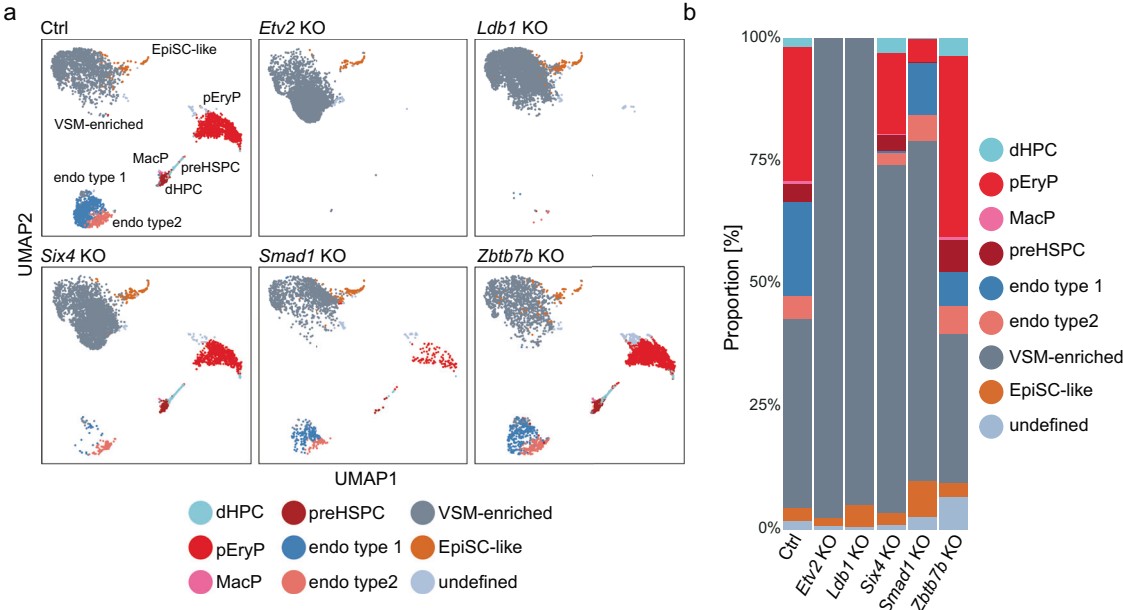

**Fig. 4 | Haematopoietic and endothelial differentiation potential in validated candidate genes. a** UMAP with overlaid FlowSOM clustering of 50,000 cells and **b** bar chart depicting frequency of cellular subsets identified by FlowSOM clustering in HB cultures started from total FLK1⁺ cells of the indicated genotypes.

EpiSC epiblast stem cell, pEryP primitive erythrocyte progenitors, preHSPC pre-haematopoietic stem progenitor cells, dHPC definitive haematopoietic progenitor cells, MacP macrophage progenitors, VSM vascular smooth muscle, endo endothelia. Source data for (**b**) are provided as a Source data file.

showed a peak of expression at the transition from DP towards F_SP mesoderm; *Smad1* gene expression peaked earlier than *Etv2*, at the beginning of DP mesoderm development but was broadly expressed in all populations; while *Ldb1*, *Six4* and *Zbtb7b* were equally expressed in the different mesodermal populations, with *Ldb1* showing the highest and *Zbtb7b* the lowest level of expression (Fig. 3i). These data show that our candidate genes (except for *Etv2*), would have been difficult to identify by employing a differential gene expression analysis between DP priMes and F_SP HEM.

In summary, by applying strict criteria to selecting genes, we were able to validate our hits with high efficiency. We confirmed the critical role of the known master regulator ETV2 in enabling F_SP HEM differentiation and identified additional factors, including SMAD1, LDB1, SIX4, and ZBTB7b, which act as drivers and repressors of F_SP HEM commitment.

### Haematopoietic and endothelial lineage commitment is differentially regulated by core TFs

We showed that the absence of the validated F_SP HEM-driving TFs led to a marked reduction of F_SP HEM cells, whereas the initial steps of primitive mesodermal differentiation, indicated by FLK1 upregulation, were not affected. On the other hand, loss of Zbtb7b led to an increased frequency of F_SP HEM cells among the FLK1⁺ cells compared to Ctrls. To further understand the downstream effect of our validated candidates on the developmental potential of the formed FLK1⁺ mesoderm, we proceeded to drive the differentiation of the corresponding KO lines towards haematopoietic and endothelial lineages. To achieve this, we sorted total FLK1⁺ cells from KO EB cultures, which contained both F_SP and DP mesodermal cells, and analysed their fate specification with our established high-dimensional flow cytometry panel. We compared these cultures to Ctrls transduced with non-targeting guides. As expected, and in line with the literature[15,32], *Etv2* KO cells had a complete block of haematopoietic and endothelial lineage commitment; only VSM-enriched and uncommitted EpiSC-like cells (CD41⁻VE-cad⁻CD71⁻Epcam⁺) formed in Etv2 KO cultures (Fig. 4a, b). *Ldb1* KO cells, similarly to Etv2 KO cells, did not differentiate to haematopoietic and endothelial cells, while contrastingly *Six4* KO cells

were still able to give rise to haematopoietic lineages, including both primitive and definitive subsets, with only a minor decrease (Fig. 4a, b). Additionally, *Six4* KO cells showed a marked differentiation block towards the endothelial lineage, especially endothelial cell type 1 cluster (Fig. 4a, b). Among all analysed validated candidate genes, fewest live cells were obtained from *Smad1* KO cultures (Supplementary Fig. 7a). Still, *Smad1* KO cells were able to form both haematopoietic and endothelial lineages, albeit at a highly reduced frequency and number (Fig. 4a, b). For endothelial lineages, similar to *Six4* KOs, *Smad1* KO cells showed mainly a reduction of the endothelial cell type 1 cluster (Fig. 4a, b). Zbtb7b KO cells resulted in the highest yield of live cells after differentiation and formed all lineages with an increase in haematopoietic lineages and endothelial cell type 2 cluster (Fig. 4a, b and Supplementary Fig. 7a). This result indicates that Zbtb7b KO mesoderm is already committed towards haematopoietic and endothelial fate and their enhanced capacity during HB culture could be a direct effect of the increased frequency of F_SP HEM cells in the starting culture. In summary, our data reveal that each identified TF has a distinct role during haematopoietic and endothelial linage differentiation, and that loss of function of those TFs leads to the formation of mesodermal subsets with a defined lineage differentiation bias.

### Heterogeneity of mesodermal cells is regulated by core TFs

Our previous differentiation experiments revealed that the developing mesoderm has distinct lineage differentiation biases depending upon individual core TF KOs. This points towards the presence of mesodermal subpopulations and cellular heterogeneity during mesoderm development. To assess this possibility and to obtain greater insight into the gene expression perturbations in the respective KO lines, we performed probe-based 10x Chromium scRNA-seq characterization of total FLK1⁺ cells in Ctrl and KO samples. Following quality control of the obtained data, we retrieved 69,174 cells (Ctrl: 17,725 cells, *Etv2* KO: 7172 cells, *Ldb1* KO: 8279 cells, *Six4* KO: 9127 cells, *Smad1* KO: 8018 cells, and *Zbtb7b* KO: 9673 cells) with a median feature count of 6054 and a median sequencing depth of 21,258 counts per cell. After optimizing the resolution parameter for Louvain clustering analysis and

conducting unsupervised dimensionality reduction and uniform manifold approximation and projection (UMAP) analysis in Seurat, we identified 10 different clusters (C) and assigned them to developmental stages (Fig. 5a).

The clusters did not form distinct, well- segregated populations, instead, distinct subpopulations are positioned next to each other, underpinning continuous developmental processes. Based on a differential gene expression analysis incorporating well-established mesodermal and haemato-endothelial genes, the developmental trajectory towards HEM could be defined (Fig. 5a, b and Supplementary Table 2) from C1-C2-C3-C4-C5-C6 (priMes1/2; Foxf1[hi] mesoderm; pre-HEM and HEM 1/2, respectively). We denoted C1 (priMes 1) as the most naïve mesodermal cluster due to high expression of early mesodermal markers such as *Eomes*, *Mesp1* and *Mixl1* (Fig. 5b). By contrast, the most advanced cluster along the HEM trajectory C6 (HEM 2) was characterised by upregulation of known haematopoietic genes crucial in the process of endothelial-to-haematopoietic transition and blood development including *Tal1*, *Gata2* and *Runx1* (Fig. 5b, c). Following our previous nomenclature of DP priMes and F_SP HEM, we assigned the clusters C1-4 to DP priMes and C5 and C6 to F_SP HEM, based on the expression of FLK1 (encoded by KDR gene) and PDGFRa (Fig. 5b, c). Combining our two scRNA-seq datasets further supports this assignment (Supplementary Fig. 7b). Strikingly, several clusters were absent in, or specific to, individual KOs and cluster frequencies are distinct across conditions: for example, Etv2 KO and Ldb1 KO cells are stalled early during development, with only C1-C3 and C1-C5 being present, respectively (Fig. 5a, d, e). This highlights the early and critical role of ETV2 during HEM development and shows that LDB1 plays a subsequent role, allowing the cells to progress further. Interestingly, *Etv2* starts to be upregulated in C4 (pre-HEM) and has its highest expression in C5 (HEM 1) (Fig. 5b, c). The early developmental block of *Etv2* and *Ldb1* KO cells agrees with the differentiation experiments where neither endothelial nor haematopoietic lineages develop in those KOs (Fig. 4a). In *Six4* KO cell cultures, all clusters were present that were in the Ctrl samples, albeit with altered frequencies, but—uniquely among the F_SP-driving TFs—*Six4* KOs were able to form cells belonging to the most advanced C6, albeit at highly reduced frequency (Fig. 5a, d, e). This is in agreement with our differentiation experiment, where *Six4* KO cells were able to form all haematopoietic lineages (Fig. 4a). Concurrently, the early C1 (primitive mesoderm1) was expanded in *Six4* KO cell cultures compared to the Ctrl (Fig. 5a, d, e). *Smad1* KO cells were the most different to the Ctrl and the other candidate KOs, forming three clusters that were almost absent in the other cultures: C7 (neuroectoderm), C8 (mixed mesoderm) C9 (early endothelial progenitors). C7 was characterised by high expression of *Nnat*, *Dll1* and *Dll3*, which are collectively expressed during neural cell development (Fig. 5b and Supplementary Table 2). C8 (mixed mesoderm) did not resemble any mesodermal population commonly found during development and was characterised by high expression of *Tll1* (Tolloid-Like Protein 1) and *Opticin* (Optc gene) (Fig. 5b and Supplementary Table 2). Alongside, C9 (early endothelial progenitors) was also mainly found in *Smad1* KO cell cultures and was defined by high expression of endothelial markers including *Plxnd1*, *Rasip1*, and *Rhoj* (Fig. 5a, b and Supplementary Table 2). Despite the different composition of mesodermal subpopulations, *Smad1* KO cells were still able to form blood and endothelial lineages albeit at highly reduced frequency (Fig. 4a). *Zbtb7b* KO cells gave rise to a higher frequency of C5 (HEM1) and a lower number of cells in C3 (Foxf1[hi] mesoderm) compared to the Ctrl (Fig. 5a−e). C5 was characterised by the highest expression of *Etv2* and additional TFs associated with haematopoietic development, including *Gata3* and *Lmo2* (Fig. 5b, c). Therefore, C5 is clearly committed towards blood development (Fig. 5b, c). C3 was characterized by expression of early mesodermal markers like *Hand1* and *Mest* and a high expression of the TF *Foxf1* (Fig. 5b, c and Supplementary Table 2). To note, a previous study implicated the expression of *Foxf1* in

inhibiting haematopoietic lineage commitment during mesoderm specification[33]. A reduced frequency in C3 or *Foxf1* expressing mesodermal cells could therefore result in the observed phenotype of enhanced F_SP HEM commitment of *Zbtb7b* KO cells and increased commitment towards the haematopoietic and endothelial lineage (Fig. 4a). The intermediate C4 (pre- HEM ) revealed only two distinct marker genes (*Lmo2* and *Fgf3*) and was present at low frequencies in all analysed conditions (ranging from 2 to 6% of the cells in the different genotypes) (Supplementary Table 2). *Zbtb7b* KO cells had the highest frequency of C4 among all analysed conditions (6.53%), pointing to an advanced commitment towards the HEM lineage (Supplementary Fig. 7c). Cells belonging to C0 did not express FLK1 but rather the pluripotency marker *Nanog*; accordingly, we denoted them as uncommitted cells which were equally present in all genotypes (between1- 3%) (Fig. 5a, b, d). Cells belonging to C0 were most likely caught during FACS sorting due to a less stringent gating strategy and were excluded from further analysis. To characterise all cells committed towards mesodermal fate we did not select for cells that only highly expressed FLK1, accepting the risk of sorting and sequencing FLK1neg cells.

To further analyse the developmental order of the clusters and to not solely rely on the expression analysis of known marker genes, we performed condition-wise pseudotime analysis (Fig. 5f and Supplementary Fig. 7d). We used C1 (priMes1) as the root, based on its naïve mesodermal transcriptomic profile. We confirmed C6 (HEM 2) as the most committed mesodermal population following the haemato-endothelial mesodermal developmental trajectory. Etv2 KO cells, *Ldb1* KO cells and *Six4* KO cells followed the same trajectory as the Ctrl cells, with the former two being stalled early during development. Intriguingly, *Zbtb7b* KO cells, which are largely composed of the same clusters as the Ctrl, formed the most advanced C6 (HEM 2) via a different trajectory, mainly through C5 (HEM 1), and included very few cells belonging to C3 (Foxf1[hi] mesoderm) (Fig. 5f). This could indicate that Zbtb7b KO cells do not form C3 or pass through it faster than the Ctrl cells. Smad1 KO cells appeared to undergo an entirely different developmental trajectory compared to the Ctrl and formed a trajectory of the Smad1 KO specific clusters via C1-C7-C8-C9 (Fig. 5f). We additionally, projected the in vitro generated cells of the individual conditions onto the in vivo haemato-endothelial trajectory graph (Fig. 1e and Supplementary Fig. 7e). In agreement with our UMAP analysis, Etv2, Ldb1 and Smad1 KOs cells display an early developmental block, whereas Six4 KO cells can be projected further onto the trajectory (Supplementary Fig. 7e). Smad1 KO cells had the lowest frequency of all analysed conditions (16.5% vs.20.4 % for Ctrl) which agrees with the alternative trajectory that *Smad1* KO cells follow (Fig. 5f and Supplementary Fig. 7e). Collectively, our scRNA-seq data of the candidate gene KOs reveals that each TF impacts distinctly on the mesodermal developmental trajectory and distinct mesodermal subpopulations form during EB cultures, which are differently impacted by candidate gene KO.

## Discussion

By applying large-scale targeted CRISPR-Cas9 KO screens in an ESC-based differentiation model of mesodermal development we identified TFs crucial for HEM specification. ETV2, the known master TF for this differentiation step, was among our top-hits and served as a positive control for the validity of our screening set-up. We additionally identified TFs which function as drivers (LDB1, SIX4, SMAD1) or repressors (ZBTB7b) of haemato-endothelial lineage commitment. Unexpectedly, our screen did not indicate any chromatin factors as major regulators of this transition, suggesting that they are not essential for HEM specification in our set-up. However, we detected several chromatin factors known to impact lineage differentiation among the top hits of depleted genes in transitions prior to priMes to HEM, including: *Ddb1*, *Kdm6a (UTX)*, *Kmt2d (MLL4)*, *Arid3b* and *Brd8*,

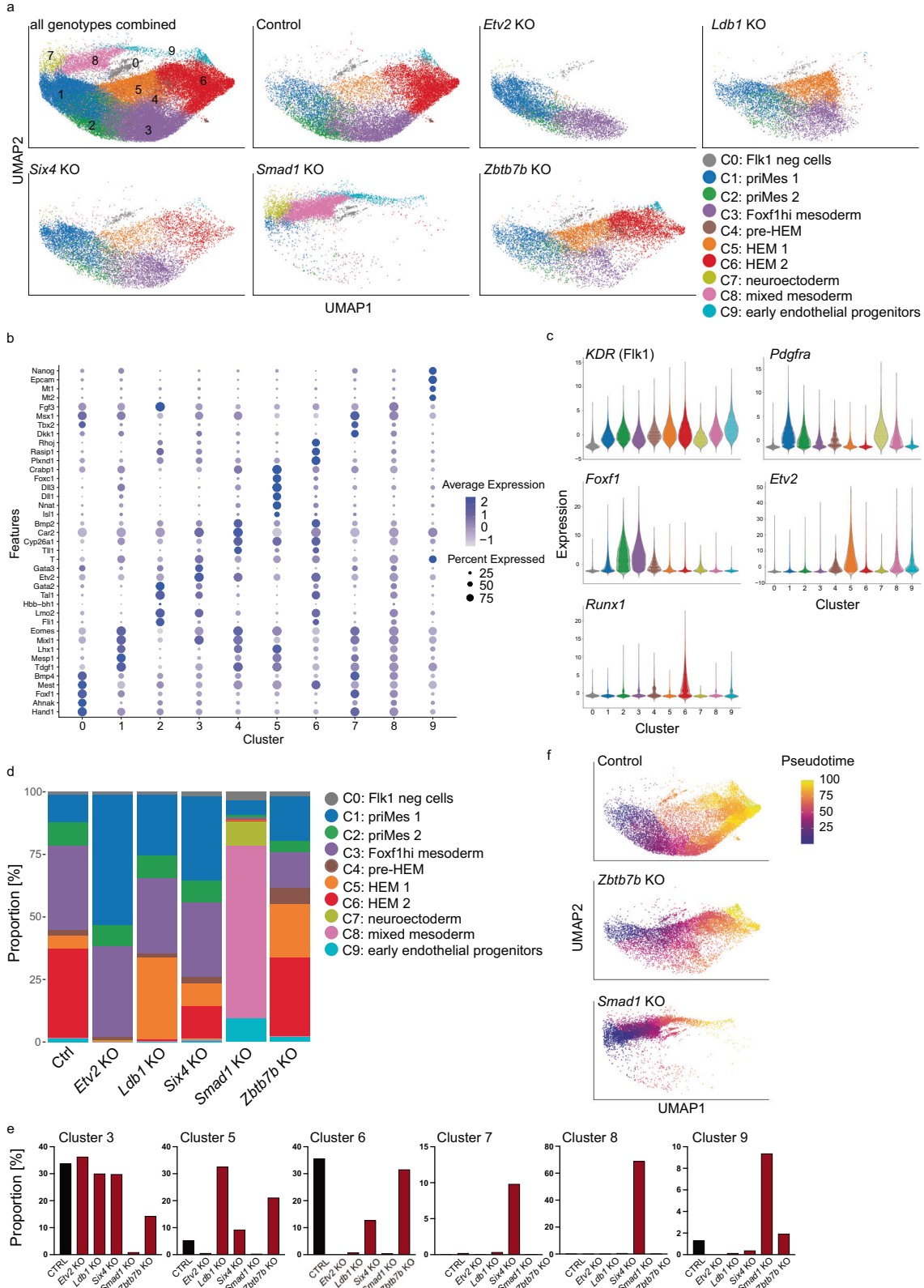

**Fig. 5 | Single-cell RNA-seq characterisation of mesodermal cells upon loss of core TFs.** scRNA-seq analysis of FLK1+ cells isolated by FACS from day 5 EB cultures for indicated genotypes **a** UMAP visualizing 10 identified clusters for all genotypes combined (upper left) or for the indicated genotypes. **b** Dot plot showing expression of signature genes among the identified clusters. **c** Violin plots showing the expression of indicated genes in the identified clusters. **d** Bar chart of proportions and **e** frequency of identified clusters in the indicated genotypes. **f** UMAP visualising the pseudotime trajectory of indicated genotypes. Source data for (**d**, **e**) are provided as a Source data file.

indicating that chromatin factors could play an earlier role along the differentiation trajectory. Therefore, the absence of chromatin regulators in the later transition of DP priMes → F_SP HEM may be attributable to our experimental design, where gene knockouts were induced at the pluripotent stage (day 0). This setup may not capture the effects of genes that act during both earlier and later stages. To better test the involvement of chromatin regulators in the DP priMes → F_SP HEM transition, an inducible screening approach could be employed, in which gene knockouts are activated at later stages of the protocol (e.g., around day 3). This would allow to: (i) more precisely test the stage-specific roles of chromatin regulators, and (ii) evaluate their contribution to the transition towards haemato-endothelial mesoderm.

The phenotypes of the validated F_SP HEM-driving TF KOs were comparable across our EB cultures, exhibiting blocked differentiation towards HEM without major effect on priMes commitment. Besides ETV2, Ldb1 and Smad1 were previously suggested as novel players in early haematopoiesis, precisely at the transition to HEM[34,35]. Our study confirms and extends these findings, thereby establishing LDB1 and SMAD1 as crucial regulators of haemato-endothelial fate specification. The gene KOs of *Etv2*, *Ldb1* and *Smad1* lead to early embryonic lethality in KO mice, which is expected for key factors of early haematopoiesis[15,36,37]. Another here-identified F_SP HEM-driving TF was Six4, which has not previously been linked to endothelial or haematopoietic development and is not essential for mouse embryogenesis[38]. The six members of the Six gene family include Six1 and Six4, whose expression patterns are almost identical; as these genes are only separated by 100 kb it is possible that they are regulated equally[39]. Six1 KO mice, contrarily to Six4 KOs, exhibit embryonic lethality with reported defects in myogenesis and multiple organs, but not the vasculature or haematopoietic system and Six1/Six4 double KOs have an augmented phenotype compared to Six1 single KO mice[40–42]. We speculate that Six1 and/or other Six family members could compensate the loss of function of Six4 in vivo. Intriguingly, Six4 KO FLK1+ cells could form blood lineages but have a marked reduction in endothelial commitment. As endothelial cells are critically needed for blood formation, it is tempting to speculate that Six4 is involved in the differentiation of haemogenic endothelial cells (HECs). Further analyses are needed to test whether cultures of Six4-expressing endothelial cells are depleted of HECs and whether Six4 has the potential to serve as a reporter gene to enrich for bona fide HECs.

We also observed major phenotypic differences between the individual KO conditions during our scRNA-seq characterisation and experiments on the differentiation of FLK1+ mesoderm. Both data sets suggested a high degree of heterogeneity within FLK1+ mesodermal cells. By relying on the two surface markers FLK1 and PDGFRα it is possible to enrich for mesodermal lineages; this approach to classification has been used exclusively in numerous publications, with a risk of oversimplification. Throughout embryonic development, the expression of these markers is mutually exclusive in mesoderm populations, with FLK1 single-positive mesoderm giving rise to extra-embryonic and lateral plate mesoderm that subsequently forms blood and vasculature, and PDGFRα single-positive mesoderm or paraxial mesoderm forming muscle and bones[5,12,43]. Several studies suggest some exceptions from this classical discrimination, for example, where PDGFRα single-positive cells have given rise to the haemato-endothelial lineage under appropriate conditions[43–45]. Still, these studies suggest that most mesodermal populations transition through the FLK1 and PDGFRα double-positive primitive mesoderm stage before they further differentiate to more committed progenitors by down-regulating or maintaining one or both markers. It is a challenge in the future to define new surface proteins that enable a finer resolution of the primitive mesodermal stage and could serve as predictors for their lineage bias.

Our single-cell transcriptomic analysis of control and KO cells revealed that loss of *Etv2* or *Ldb1* led to an early arrest during mesodermal development, whereas *Smad1* KO cells follow an entirely different trajectory. Due to the similar phenotypes of Etv2 and Ldb1 KO cells, it is possible that both factors are acting together to allow HEM commitment. SMAD1 is the signal transducer of bone morphogenetic protein (BMP) subgroup of TGFβ molecules[46]. BMP signals are known to be important regulators of early embryonic development; more precisely, BMP4 is a crucial cytokine needed for mesodermal development in both mice and humans[47,48]. Loss of SMAD1 had the strongest effect on the composition of mesodermal subpopulations, but unexpectedly, some cells were still able to differentiate into mesoderm that gave rise to both haematopoietic and endothelial lineages. We speculate that, upon lack of Smad1 the differentiating cells do not perceive the appropriate developmental signals driven by BMP4 and start differentiating towards the default programme of ESCs, acquiring undefined or neural signatures. We also identified the zinc finger protein ZBTB7b as a repressor of priMes differentiation. *Zbtb7b*, also known as *ThPok*, functions in T cell development as a master regulator of CD4/CD8 lineage determination in the thymus[49,50], but has not yet been implicated in early blood development. It was shown recently that *Zbtb7b* regulates the primed-to-naïve transition (PNT) of pluripotent stem cells after BMP4 administration in a model of EpiSC-to-ESC conversion[51]. *Zbtb7b*, together with *Zbtb7a* were reported to facilitate opening of naive pluripotent chromatin loci and the subsequent activation of nearby genes[51]. Our scRNA-seq data revealed that loss of *Zbtb7b* led to a different mesodermal developmental trajectory, where one cluster characterised by high Foxf1 expression is markedly reduced. Foxf1 was previously shown to inhibit mesodermal development towards a haemato-endothelial fate and reduction of Foxf1hi mesoderm could indeed lead to an enhanced potential to commit towards the haemato-endothelial lineage[33]. Future research will address if *Zbtb7b*, as suggested by the PNT study, is involved in chromatin remodelling during mesodermal differentiation; if so, this could impact the composition of mesodermal subpopulations and establishment of a chromatin landscape favourable to haemato-endothelial differentiation.

Importantly, despite our efforts to analyse the gene expression perturbations resulting from loss-of-function of all the validated candidates, mechanistic understanding of the gene function and of the interplay between the identified TFs is needed in the future. Furthermore, our approach of modelling early haematopoiesis ex vivo employing murine ESCs enabled us to obtain sufficient cell numbers for large-scale genetic screenings with the optimal sgRNA representation during a delicate developmental time window that would have been challenging, if not impossible in vivo. Still, our study would be strengthened by in vivo characterisations and experiments addressing the function of the identified core TFs during early mesodermal and blood commitment in mouse development. For these in vivo experiments, factors that lead to early embryonic lethality, like Etv2, Ldb1 and Smad1 would have to be contrasted to factors that have not been implicated in early haematopoiesis or have been associated with impaired survival, like Six4 and Zbtb7b.

In conclusion, our study identified factors involved in HEM differentiation that shape the developmental potential of primitive mesodermal cells. These findings suggest a previously unappreciated complexity in the interplay between TFs regulating mesodermal fate specification and reveal several key players in embryonic haematopoiesis. Not only do these insights fill important gaps in our knowledge of this process, but this work will pave the way for much-needed improvements to stem cell derivation/expansion protocols, with the potential to advance clinical applications of these cells for the treatment of blood diseases.

## Methods

### Cell culture and cell line generation

Mouse ESCs (HA36CB1, 129×C57BL/6) were cultivated on 0.2%-gelatine-coated dishes in Dulbecco's Modified Eagle Medium (DMEM) supplemented with 15% foetal calf serum (FCS), 1x non-essential amino acids, 1 mM L-glutamine, LIF, and 0.001% β-mercaptoethanol.

The KO cell lines were generated by co-transfecting pX330-U6-Chimeric_BB-CBh-hSpCas9 (Addgene 42230) with two sgRNAs (sequences see Supplementary Table 3). Control samples were transfected with non-targeting sgRNAs (sgRNA1_GCGTATCTACCCTAC CGCCG, sgRNA2_GCGGTTACCGCGAAAACCAT). pRR-Puro recombination reporter[52] (Addgene 65853) was co-transfected, and after 36 h, cells were treated with 2 µg/ml of puromycin for another 36 h. KO clones were validated by Sanger sequencing. KO validation by western blot detection was performed as described previously[53] for the identified candidates Etv2, Ldb1, Six4, Smad1 and Zbtb7b (Supplementary Fig. 8a–e). Transfections were conducted using Lipofectamine 3000 reagent (Thermo Fisher Scientific) at a 2:1 Lipofectamine/DNA ratio in OptiMEM (Thermo Fisher Scientific).

### Embryoid body (EB) culture for mesoderm differentiation

Twenty-four hours prior to the initiation of EB cultures, ESCs were transferred into gelatinized plates containing Iscove's Modified Eagle Medium (IMDM) (Gibco). For the generation of EB cultures, ESCs were trypsinized then plated at 10,000 cells/ml in non-adherent 10 cm$^2$ petri dishes in EB medium (IMDM supplemented with 1% L-glutamine (Gibco), 10% FCS (Gibco), 0.6% transferrin (Roche, 10652), 50 µg/ml ascorbic acid (Sigma, A4544) and 0.03% monothioglycerol (Sigma, M6145). BMP4 was added from day 0 of EB culture, and bFGF, activin A and VEGF from day 2.5 (all Peprotech) all at 5 ng/ml. After 5 days in culture, EBs were harvested, and TrypLE Express Enzyme (1×) (Gibco, 12605036) was used to generate a single-cell suspension.

### Haemangioblast (HB) culture for haematopoietic and endothelial commitment

Prior to HB culture, mesodermal populations underwent FACS sorting (for either DN, P_SP, DP priMes, F_SP HEM or total FLK1+) and $0.5 \times 10^6$ cells were cultured on 0.2%-gelatine-coated plates in IMDM supplemented with 1% penicillin–streptomycin, 1% L-glutamine, 15% FCS (Gibco), 0.06% transferrin (Roche), 0.03% monothioglycerol (Sigma, M6145), 50 mg/µL of ascorbic acid (Sigma, A4544), 5 ng/ml VEGF (Peprotech) and 10 ng/mL IL-6 (Peprotech). After 2.5 days in culture, cells were harvested for flow cytometry analysis. TrypLE Express Enzyme (1×) (Gibco, 12605036) was used to generate a single-cell suspension.

### Long-term haematopoietic assay in liquid media

Defined numbers (18,000, 10,000, or 5000 cells) of FACS-sorted F_SP HEM or DP priMes were plated on 0.2%-gelatine-coated 96-well plates. For each cell number a minimum of 24 wells were seeded. The cells were first cultured for 2.5 days in HB culture media (to induce the differentiation to haematopoietic progenitors) and subsequently in conditions supporting the maturation of haematopoietic cells in IMDM supplemented with 1% L-glutamine, 10% FCS (Gibco), 0.6% transferrin (Roche), 0.03% monothioglycerol (Sigma, M6145), 50 mg/µL of ascorbic acid (Sigma, A4544), 100 ng/ml SCF (Peprotech), 1 ng/ml IL-3 (Peprotech), 10 µg/ml G-CSF (Peprotech), 5 µg/ml IL-11 (Peprotech), 5 µg/ml IL-6 (Peprotech), 5 µg/ml TPO (Peprotech), 10 µg/ml M-CSF (Peprotech) and 2000 U/ml erythropoietin (R&D Systems). After 9.5 days in culture wells were scored for the presence of haematopoietic cells and cells were harvested for flow cytometry analysis. TrypLE Express Enzyme (1×) (Gibco, 12605036) was used to generate a single-cell suspension.

### Flow cytometry

**EB cultures.** Single-cell suspensions were obtained on day 5 of differentiation. For cell-surface labelling, cells were incubated for 30 min at 4 °C with a 1:200 solution of anti-CD309 (FLK1) (APC, BD Bioscience, clone Avas12a1) and anti-CD140a (PDGFRα) (PE, BD Bioscience, clone APA5) monoclonal antibodies. LIVE/DEAD Fixable Near-IR Dead Cell Stain (L34975, Invitrogen) was used to determine cell viability. Sample data were acquired using a BD Symphony (BD Biosciences) flow cytometer, and data were analysed using FlowJo software (version 10.7, Tree Star), then visualized with Prism (version 5.0a).

**Haemangioblast (HB) cultures.** Single-cell suspensions were obtained at day 2.5 of differentiation.

*Conventional FACS analysis:* Antibody staining was performed for 30 min at 4 °C with a 1:200 solution of anti-CD41 (BV650, BD Biosciences, Clone MWReg30) anti-VE-cadherin (eFluor 660, BD Biosciences, clone eBioBV13), and anti-cKit (PE-Cyanine7, BD Biosciences, clone ACK2) monoclonal antibodies using a BD Symphony (BD Biosciences) flow cytometer. LIVE/DEAD Fixable Near-IR Dead Cell Stain (L34975, Invitrogen) was used to determine cell viability.

*Spectral flow cytometry:* Cells were first labelled with Zombie NIR fixable viability dye (BioLegend, 1:500) for the exclusion of dead cells. Before antibody staining, cells were incubated with TrueStain FcX™ and TrueStain Monocyte Blocker (BioLegend) to reduce nonspecific binding. Surface antigens were labelled using an antibody panel consisting of the following 23-markers (Supplementary Table 1): From BD biosciences: anti-CD117(cKit) (BUV395, Clone 2B8), anti-CD43 (BUV496, Clone 1B11), anti-CD45 (BUV563, Clone 30-F11), anti-CD9 (BUV661, Clone KMC8), anti-CD44 (BUV737, Clone IM7), anti-CD31 (BUV805, Clone 390), anti-Epcam (CD326) (BV480, Clone G8.8), anti-CD93 (BV605, Clone AA4.1), anti-CD41 (BV650, Clone MWReg30), anti-CD14 (FITC, Clone rmC5-3), anti-CD71 (RB780, Clone C2), and anti-CD47 (PE-CF594, Clone miap301). From BioLegend: anti-CD64 (BV421, Clone X54-5/7.1), anti-CD16/32 (BV711, Clone 93), anti-CX3CR1 (BV785, Clone SA011F11), anti-CD202b (Tie2) (PE, Clone TEK4), anti-CD34 (PE-Cy5, Clone MEC14.7), anti-CD309 (FLK1) (PE-Cy7, Clone Avas12), anti-Ly6C (Alexa Fluor 700, Clone HK1.4), anti-Sca-1 (APC-Fire 750, Clone D7), and anti-CD11b (APC-Fire 810, Clone M1/70). From ThermoFisher: anti-CD11c (PE-Cy5.5, Clone N418), and anti-VE-cadherin (CD144) (eFluor660, Clone BV13). Surface staining took place on ice for 20 min in phosphate-buffered saline (PBS), followed by three PBS washes. Cells were subsequently acquired using a Cytek Aurora 5L spectral flow cytometer (Cytek Biosciences).

*Data preprocessing and export:* Doublets were excluded based on side scatter area versus height, and non-viable cells identified by Zombie NIR™ positivity were removed using FlowJo (v10, TreeStar). The resulting compensated, pre-gated dataset was exported in FCS format for further analysis with R software (version 4.1.2).

*Transformation and normalization of data:* The flowCore R package was utilized to import the FCS files via the read.flowSet() function. Fluorescence intensity values were transformed using an arcsinh function to address the wide dynamic range. To facilitate direct comparisons across samples, data underwent percentile normalization, scaling expression values to a consistent 0–1 range.

*Dimensional reduction and clustering:* All 23 markers, including lineage- and progenitor-associated proteins, were subjected to dimensional reduction through uniform manifold approximation and projection (UMAP), implemented via the umap R package. For unbiased identification of cellular subsets, FlowSOM clustering was conducted using the same transformed and normalized data. FlowSOM-derived clusters were grouped through metaclustering, with subsequent refinement by manual annotation guided by expert assessment. Cluster identities were determined based on median marker expression levels, merging clusters exhibiting similar marker profiles. This approach defined clear subsets of primitive and definitive

haematopoietic progenitor cells, endothelia, and VSM-like cells. A dot plot depicting median marker expression per cluster is shown in Fig. 1g to illustrate cluster characterization.

**Long-term haematopoietic assay in liquid media.** Single-cell suspensions were obtained on day 9.5 of differentiation. For cell-surface labelling, cells were incubated for 30 min at 4 °C with a 1:200 solution of anti-CD45 (PerCP-Cyanine 5.5, BD Biosciences, Clone 3D-F11) and anti-CD71 (RB780, BD Biosciences Clone C2) monoclonal antibodies. LIVE/DEAD Fixable Near-IR Dead Cell Stain (L34975, Invitrogen) was used to determine cell viability. Sample data were acquired using a BD Symphony (BD Biosciences) flow cytometer, and data were analysed using FlowJo software (version 10.7, Tree Star), then visualized with Prism (version 5.0a).

**Poly-A RNA-sequencing of EB cultures on day 5.** Viable $0.5 \times 10^6$ cells of DN, P_SP, DP priMEs, and F_SP HEM were FACS sorted on day 5 of EB differentiation, using FLK and PDGFRa surface expression. Subsequently, RNA was isolated using Qiagen RNAeasy kit (Qiagen cat. 74134). The quality of the isolated RNA was determined with a Fragment Analyzer (Agilent, Santa Clara, California, USA). The TruSeq Stranded mRNA protocol (Illumina, Inc., California, USA) was used in the succeeding steps. Briefly, total RNA samples (100–1000 ng) were poly A enriched and then reverse-transcribed into double-stranded cDNA. The cDNA samples were fragmented, end-repaired, and adenylated before ligation of TruSeq adapters containing unique dual indices (UDI) for multiplexing. Fragments containing TruSeq adapters on both ends were selectively enriched with PCR. The quality and quantity of the enriched libraries were validated using the Fragment Analyzer. The libraries were normalized to 10 nM in Tris-Cl 10 mM, pH8.5 with 0.1% Tween 20. Single-end 100 bp were sequenced on Novaseq 6000 (Illumina, Inc., California, USA), and an average of ~20 million reads per sample were obtained.

*Data Analysis:* Reads were mapped to the GRCm39.107 genomic reference and count tables generated using STAR-2.7.7[54]. Count tables were filtered and only genes with a minimum of 10 reads and occurrences in at least three conditions were retained (24155genes). PCA was conducted using DESeq2 v1.38.3 R package[55].

**Single-cell sequencing.** SORT-seq of EB cultures day 5 and HB cultures day 2.5

Viable single cells underwent single-cell FACS-sorting into 384-well cell capture plates (Single Cell Discoveries, Netherlands) using Flk1, CD41, c-kit, and VE-cad surface expression (gating strategy in Supplementary Fig. 4a). Each well contained 50 nl of barcoded primers and 10 μl of mineral oil (Sigma M8410). After sorting, plates containing cells were centrifuged at $1500 \times g$ for 1 min then placed on dry ice before storage at −80 °C. Single-cell RNA sequencing was performed by Single Cell Discoveries according to an adapted version of the SORT-seq protocol[56], with primers as described in ref. 57. Briefly, cells were heat-lysed at 65 °C followed by cDNA synthesis, before the barcoded material from each plate was pooled into its own library and amplified by in vitro transcription (IVT). Library preparation was then carried out following the CEL-Seq2 protocol[58] to generate a cDNA library for sequencing using TruSeq small RNA primers (Illumina). The DNA library was paired-end sequenced on an Illumina Nextseq 500, high output, with a 1 × 75 bp Illumina kit (read 1: 26 cycles, index read: 6 cycles, read 2: 60 cycles).

*Data analysis:* During sequencing, read 1 was assigned 26 base pairs and was used to identify the Illumina library barcode, cell barcode, and unique molecular identifier (UMI). Read 2 was assigned 60 base pairs and used to map to the reference genome Mus musculus (GRCm38) version 99 with STARSolo 2.7.3a. Briefly, mapping and generation of count tables were automated using the STARSolo 2.7.3a aligner. Unsupervised clustering and differential gene expression

analysis was performed with the Seurat5 R toolkit. Count tables were imported into R, where a singleCellExperiment object was constructed and subsequently filtered based on library size, feature count, spike-in count, and mitochondrial count using the isOutlier function from the scater v1.26.1 R package. Following quality control, the dataset was analysed using the Seurat v4.3 R package, using the SCTransform workflow[59]. This included normalization, scaling, and linear dimensionality reduction. Integration of selected haemato-endothelial cells from the publicly available mouse gastrulation atlas[27] with our in vitro scRNA-seq dataset was performed using Seurat. To complement this integration, the force-directed graph of the haemato-endothelial landscape was reconstructed by incorporating supplementary data from the atlas publication. Within the integrated dataset, in vitro population labels were assigned to the in vivo cells from the atlas based on the $k = 5$ nearest neighbours. For RNA velocity analysis, the Seurat data were exported, and an annotationData object was manually constructed in Python. Subsequently, scvelo v0.2.5 (ref. 60) was applied to it for comprehensive RNA velocity analysis. The latent time of each cell from the dynamical model was used to calculate an average per cluster, thereby arranging the clusters along the developmental trajectory.

**10x chromium fixed RNA profiling.** Isolated cells were fixed using the 10x Chromium Next GEM Single Cell Fixed RNA Sample Preparation Kit (PN-1000414) and stored at −80 °C. The quantity of fixed cell suspension was assessed using an automated cell counter (LUNA-FX7, Logos). Library preparation was conducted using the 10x Chromium Fixed RNA Profiling Reagent Kits for Multiplexed Samples. Briefly, four fixed samples underwent probe hybridization, pooling, and loading onto the chip. Approximately 66,000 cells from the sample pools were loaded into the 10x Chromium X. Library preparation followed the manufacturer's guidelines (Guide CG000527, Rev E). For sequencing, the resulting libraries were processed on two lanes of an Illumina NovaSeq X Plus 10 billion flow cell with a 150 bp paired-end read configuration. An average sequencing depth of around 25,000 reads per cell was achieved.

*Data analysis:* Single-cell RNA-seq reads were processed with the 10x software CellRanger Multi version 7.2.0[61] with the Mus Musculus GENCODE GRCm39 annotation (Release M31-2023-01-30) to generate the single-cell gene expression matrix. Count tables were imported into R using the Read10X function of Seurat v4. A Seurat object was created and filtered based on library size, feature count, and mitochondrial count, using the isOutlier function of the scater v1.26.1 R package[62]. Normalisation, scaling, and variable feature detection were performed using the SCTransform workflow in Seurat. Subsequently, linear dimensionality scaling, unsupervised clustering and differential gene expression analysis/positive marker discovery were performed in Seurat. Integration with other scRNA-seq datasets was performed using Seurat functions SelectIntegrationFeatures, PrepSCTIntegration, FindIntegrationAnchors, and IntegrateData with SCT as normalisation method. For the MouseGastrulationAtlas in vivo dataset[27], condition-wise integration was conducted on (per-condition) down-sampled subsets. Within the mixed dataset, labels have been propagated to the $k = 3$ nearest neighbours. During integration with our previous in vitro EB dataset (sort-seq) a single per-cluster down-sampled dataset was used. For pseudo-time analysis, the Seurat object was exported to a CDS object using the SeuratWrappers package. Pseudotime trajectory analysis was conducted using Monocle3[63].

The trajectory graph was constructed on the previously established UMAP embeddings from Seurat via the "learn_graph" function, using default options, but "use_partition" being set to false. Cluster 3 cells were designated as the root population based on their gene expression profile, indicating a more immature state. Cells were ordered along this trajectory using the "order_cells" function to establish pseudotime coordinates.

**CRISPR screens in embryoid body EB cultures.** Lentiviral plasmid library targeting TFs and chromatin-regulators (EpiTF library) was kindly provided by Michlits et al.[64]. Three independent CRISPR screens were performed with a guide representation of ~ 500x. For lentiviral transduction $60 \times 10^6$ ESCs that constitutively expressed Cas9[28] were spinfected at a multiplicity of infection (MOI) of 0.2 with polybrene, by centrifugation $500 \times g$ for one hour, at 37 °C. After spinfection, cells were incubated at 37 °C, 5% $CO_2$ for 48 h and subsequently underwent FACS based on GFP expression to isolate sgRNA+ cells. Per screen, $12 \times 10^6$ sorted sgRNA+ cells were used for EB differentiation. Mesodermal populations of interest underwent FACS on day 5 based on FLK1 and PDGFRα expression. A timepoint zero sample (T0) was collected at the start of the cultures.

*CRISPR screen sequencing library preparation and sequencing:* Genomic DNA was isolated with the DNeasy Blood 7 Tissue Kit (Qiagen) following the manufacturer's instructions. DNA concentrations were measured using the Qubit dsDNA BR assay, and the entire samples were used for the first PCR amplification. In PCR1, sgRNAs were amplified and equipped with adapters using Herculase II Fusion DNA Polymerase (Agilent, 600677). Following, PCR products were pooled and cleaned up by Qiagen MinElute PCR Purification kit following manufacturer's instructions (Qiagen) and subsequently size selected with 0.7× AmpureXP beads, according to the manufacturer's instructions. DNA concentration of cleaned-up PCR products was calculated by Qubit dsDNA HS assay. In PCR2, NEBnext dual indices (NEB) were added to the amplicons using NEBNext Ultra™ II Q5 Master Mix (NEB). Following, PCR products were pooled and cleaned up by Qiagen MinElute PCR Purification kit following manufacturer's instruction (Qiagen) and subsequently size selected with 0.7× AmpureXP beads, according to the manufacturer's instructions and analysed on an Aligent TapeStation 2000 using High Sensitivity D1000 screen tape. Exact library quantification was carried out via qPCR using the KAPA Library Quantification Kit (Roche). Finally, an equimolar pool of all samples was established and sequenced on a NovaSeq 6000 sequencing machine with 100 bp single read configuration.

*CRISPR screens data processing:* Basecalling and demultiplexing were performed using bcl2fastq2 v2.20.0.422 (Illumina), and sgRNA-sequences were isolated from the reads using cutadapt v3.1 via hard-clipping. Next, sgRNA-sequences were aligned to the library using bowtie2 v2.3.5 and a counttable was assembled. Finally, sgRNA abundances were ranked on the gene-level using the MAGeCK v0.5.9 test module[65]. Rankings were then integrated by calculating the mean rank, and *p* values were combined by Fisher's method. Duplicate mean ranks were resolved by additional sub-ranking based on the best rankings across all screens.

## Statistics and reproducibility

No statistical method was used to pre-determine sample size. All experiments were confirmed by replication at least once. For each generated KO line, three independent clones were generated. The exact handling of replicates is depicted in figure panels. Randomization/Blinding is not relevant for the experiments performed in this study. Samples were allocated to either wild type control or mutant and processed in parallel through identical analysis pipelines.

## Reporting summary

Further information on research design is available in the Nature Portfolio Reporting Summary linked to this article.

## Data availability

Bulk and scRNA-seq data sets generated in this study have been deposited to NCBI GEO under the accession numbers GSE302791, GSE287036, and GSE302880, respectively. Source data are provided with this paper.

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

## Acknowledgements

We thank F. Caiado (University Hospital Zurich) for discussions and advice. We thank members of the Schmolka and Baubec laboratory for

their input and criticism. We thank Basak Corak (UZH) for technical advice. Furthermore, we thank members of the Functional Genomics Centre Zurich and members of the Cytometry Facility at the UZH for their support. We thank Single Cell Discoveries for their single-cell sequencing services. Funding: This work was supported by a post-doc fellowship of the University of Zurich, University of Zurich Research Talent Development Fund Award (FAN), Swiss National Science Foundation Ambizione grant (186012) and European Union – ERC-PT-A- Projects of the Portuguese Foundation for Science and Technology all to N.S.

## Author contributions

N.S.: Conceptualization, Formal analysis, Supervision, Investigation, Visualization, Methodology, Project administration, Writing-original draft, Writing-review and editing. M.T.: Formal analysis, Investigation, Visualization, Writing-review and editing; T.W.: Formal analysis, Investigation, Visualization; S.B.: Investigation; P.Z.: Investigation; I.M.: Formal analysis, Writing-review and editing; S.N.: Investigation, C.M.: Investigation, U.E.: Investigation. C.L. Investigation, Writing—review and editing; B.B.: Investigation; A.R.G.: Investigation, Supervision, Writing-review and editing; T.B.: Investigation, Supervision, Writing-review and editing.

## Competing interests

The authors declare no competing interests.
