## [Transparent Peer Review file · Nature Communications]

Targeted CRISPR-Cas9 screening identifies core transcription factors controlling murine haemato-endothelial fate commitment

Corresponding Author: Dr Nina Schmolka

Version 0:

Reviewer comments:

Reviewer #1

(Remarks to the Author)

Overview: This manuscript reports a novel transcription factor network that drives the transition of murine primitive mesoderm to hemato-endothelial mesoderm (HEM)—extending previous work that has identified Runx1, Gata2 (and more recently, ETV2) as key regulators of early blood lineage commitment. Using a CRISPR-Cas9 knockout screen that targets TFs in a mouse ESC blood differentiation model, the authors identify four novel transcriptional regulators (Ldb1, Smad1, Six4, and Zbtb7b) that control lineage specification from primitive toward hemato-endothelial mesoderm. Teske and colleagues subsequently perform loss-of-function studies by knocking out each TF individually, which perturbs normal hematopoiesis in vitro, thus validating these genes as functional regulators of hemato-endothelial lineage commitment. Taken together, this technically rigorous screen resulting in identification of a novel TF network crucial for HE commitment advances our understanding of the complex networked regulators of murine hematopoiesis, which will inform ongoing and future efforts to generate hematopoietic cells from pluripotent stem cells (in particular, because generation of specific blood lineages hinges upon modulation of distinctly patterned subsets of mesoderm at the earliest stages of embryonic development).

Summary of study: Teske and colleagues first characterize and validate an in vitro EB-culture mESC blood differentiation model, based off prior protocols by Gordon Keller's and Georges Lacaud's groups. They describe the progressive formation of distinct mesodermal entities defined by FLK1 and PDGFR, where cells progress from populations that are (1) double-negative to (2) PDGFR single-positive to (3) double-positive to (4) FLK1 single-positive, consistent with mesodermal commitment. RNA-seq reveals progressive downregulation of pluripotency markers and upregulation of mesodermal and hemato-endothelial genes along this developmental trajectory, while FLK1-single positive populations exhibit increased developmental potential for blood lineages (assayed for CD41 and VE-Cadherin).

Next, the authors perform a pooled CRISPR KO screen to identify putative drivers and repressors of hemato-endothelial (HEM) commitment and recover a list of candidate regulators of the transition from primitive mesoderm (double positive) toward FLK1-single positive HEM, identifying top depleted and enriched genes. The authors subsequently generate independent KO ES cell lines and differentiate them to assess relative competence to generate Flk1+ single-positive mesoderm, validating Smad1, Ldb1, Six4 as drivers—and Zbtb7b as a repressor—of the HE mesodermal commitment from primitive mesoderm. ETV2 was recovered as a top depleted hit and confirmed as a driver of primitive-to-HE mesoderm transition in accordance with previous studies, serving as a positive control.

To further uncover the effects of each gene KO on the downstream differentiation potential of HE mesoderm, the authors drive mesoderm cells toward hematopoietic and endothelial lineages, followed by analysis of fate specification, revealing that loss of function of each TF significantly biases the lineage differentiation of mesoderm subsets. Finally, the authors perform scRNA-seq characterization of FLK1+ cells in each KO line and identify 10 clusters corresponding to different developmental timepoints, finding that several clusters are absent in or specific to individual TF KOs, and that frequencies are distinct across genetic perturbations. Through pseudotime analysis, they construct a mesodermal developmental trajectory and demonstrate that each regulator TF distinctly modifies the trajectory of lineage commitment.

Major points:

1. The screen is performed on the basis of loss of PDGFR expression in FLK1+ cells as demarcating the transition from

primitive to hemato-endothelial mesoderm. While the authors do show in Figure 1c/d that Flk1+ single-positive cells are more competent to form VeCad+ CD41+ cells (and to give rise to hematopoietic and endothelial lineages), there are still significant numbers of cells that do not induce VeCad/CD41 expression at EB Day 5, suggesting that the Flk1 SP population is heterogeneous. The overall significance of the screen could be strengthened by molecular characterization of more mature blood cells differentiated from the FLK1+PDGFR - (F_SP HEM) versus FLK1+PDGFR + (DP priMes) fractions respectively, beyond merely showing that the SP fraction generates slightly higher percentages of VeCad+ and CD41+ cells. While the bulk and scRNA-seq data on EB Day 5 cells is convincing in demonstrating that early mesodermal markers and hematopoietic and endothelial markers are both segregated and enriched in DP_SP and F_SP HEM cells respectively, further functional and molecular characterization of specific blood cell types descending from these fractions of mesoderm (beyond solely flow cytometry analysis of surface markers in Fig 1e at 2.5 days post-EB culture) would reinforce the biological significance of PDGFR downregulation. (For example, demonstration of enhanced emergence of HSCs or of other cell types, from the VeCad+CD41+ fraction arising from F_SP as compared to DP preMes (or even from the bulk F_SP HEM population as compared to DP priMes) would provide further assurance that PDGFR downregulation does serve as a developmentally relevant transition.)

2. In this study, characterization of in vitro differentiated hemato-endothelial lineages (HB culture) is performed at 2.5 days past FACS isolation of Day 5 EBs. The developmental significance of the authors' findings could be significantly strengthened if they examined functional effects of modulating any of the four TFs on blood cells further differentiated down their developmental trajectory, which would lend credence to biological and functional implications of perturbing early mesoderm patterning for hematopoiesis. (For example, some functional effects of Zbtb7b KO on EB-derived cells after more than 2.5 days of HB culture would be interesting, and it would be especially of relevance to understand if knock-out of Zbtb7b might skew differentiation or enhance the production of certain more mature blood lineages.) Moreover, because there remains incomplete understanding of how faithful the in vitro differentiation of murine ES cells mimics hematopoiesis in vivo, a limitation of the study is validation of the TFs in the context of murine hematopoiesis. While generating knockouts and evaluating hematopoiesis in murine embryos is beyond the scope of this study, some discussion of the limitations of the study for revealing in vivo mechanisms of hematopoiesis is warranted.

3. While not necessarily a pivotal issue with regard to the validity of the screen, it would strengthen the claim that each TF is specifically a negative or positive regulator of mesodermal commitment if authors could demonstrate that at least one of the KO phenotypes could be rescued by reintroducing the missing TFs exogenously through a lentiviral vector, etc.

Minor points:

1. While the authors quantified sgRNAs in the T0 ESCs and were thus able to identify differences in guide abundance between DP and FLK1+ single-positive mesoderm, it would be informative and strengthen the robustness of this specific screening platform to show that sgRNA representation in the transduced mESCs (post-selection for Cas9-GFP+ cells) mirrors the representation of the EpiTF plasmid library, which would demonstrate that sgRNAs were introduced into ESCs without significant biases (e.g., a simple Lorenz curve in the supplement to show sgRNAs are uniformly expressed in ESCs without biases).

2. Typo on page 9, line 234: "but was broadly expressed in all populations; while Ldb1, Six4."

3. Typo on page 11, line 346: "sort due to a less stringent gating strategy."

4. Page 11, line 242 "Supp Fig 5b" should instead be "Supp Figure 4b."

Reviewer #2

(Remarks to the Author)

In this manuscript, Teske et al. describe the identification of novel transcription factors involved in specifying hematovascular mesoderm (HVM). Using CRISPR/Cas9 screening, authors revealed that Smad1, Ldb1, Six4, and Zbtb7b had a major impact on the formation of Flk1 single positive HVM. Results from CRISPR screening were confirmed using corresponding KO lines. Although significant research efforts have been devoted to studying factors involved in the formation of hemogenic endothelium and endothelial-to-hematopoietic transition (EHT), the initial steps in mesodermal specification remain understudied. Thus, the findings are novel and significant. I have only minor comments, mostly related to some data interpretation.

1. To assess the impact on commitment versus viability, the authors evaluated proliferative potential and viability after gene knockout. What about the possibility that certain genes might delay the maturation of the Flk1 SP population instead of abrogating it?

2. There is some confusion regarding the phenotype used to identify the cell populations defined in the main and supplementary Fig. 1. The VSM population appears to express very high levels of endothelial/early hematopoietic markers CD43, CD44, and CD202b. This could be due to issues with cell separation by flow or the presence of the CD31+CD144- endothelial population in cultures.

3. C8 and C7 may represent different parts (anterior-posterior) of the primitive streak based on the high expression of MIXL1, EOMES, MESP1, LHX1, and TLL1 genes found in the primitive streak. It could be useful to check/show additional primitive streak markers, Meox, GATA6, SOX17 and FOXA2, to see if these populations reflect different parts of the primitive streak. C2 and C3 most likely represent a lateral plate mesoderm since cells express very low levels of T and high levels of the lateral plate mesoderm markers FOXF1, HAND1, and BMP4.

4. I would discourage using the term "mature endothelial cells" to define C9 cluster cells. These emerging endothelial cells

co-express KDR and likely lack arterial and venous markers, thus representing early endothelial progenitors or emerging endothelial cells rather than mature endothelial cells. They also express high levels of the PLXND1 gene, which is involved in cardiovascular development, and could be endothelial cells developing from the anterior primitive streak committed to cardiac fate.

5. Table 1 is in supplemental materials and, therefore, should be labeled as Table S1. Otherwise, it is challenging to locate this Table.

Reviewer #3

(Remarks to the Author)

In this manuscript, the authors perform a CRISPR-Cas9 KO screen in a murine embryonic stem cell (ES) system to identify novel regulators of haemato-endothelial mesoderm differentiation. Upon identifying potentially interesting candidates in their screen, they apply selection criteria to restrict the number of hits down to 4 genes and proceed to generate stable KO ES cell lines. They next combine spectral and classic flow cytometry with single cell RNA sequencing to characterize the effect of genetic deletion of the hit genes during haemato-endothelial differentiation.

The methodologies in this study are generally solid and the experiments appear to be well designed and executed (although lacking in reporting detail – see below). However, I am not convinced that the data reported in this manuscript can address the main questions that the authors would like to answer. Quoting from the introduction, lines 88-90: “The major open question in the field today is how these - and possibly other yet unidentified - regulators cooperate to allow the initial transition towards the blood lineage by the differentiation of priMes to the HEM”. Although some transcription factors potentially involved in haemato-mesoderm formation are identified here, there is no data about how these genes cooperate with existing factors, as only single KO lines are analyzed; no molecular studies are performed to evaluate how the identified TFs interact. Furthermore, I find the title of the MS “Targeted CRISPR-Cas9 screening identifies transcription factor network controlling murine haemato-endothelial fate commitment” misleading as in this manuscript no analysis of transcription factor networks is provided.

Of the four validated genes, two were previously known as regulators of hemato-endothelial differentiation (Ldb1 and Smad1, refs. 35 and 36 and others). The other gene validated as a putative positive regulator, Six4, was not previously implied in HEC differentiation, but it appears not essential for blood generation in their system (See Fig. 3b, 4a) though its KO results in a decrease in the frequency of Flk1+ mesoderm. At the same time, the KO of Zbtb7b, putative repressor of hemato-endothelial differentiation, only results in an increase of primitive erythrocytes (Figure 4), a lineage not necessarily derived by HEC, and does not result in an increase of Runx1+ HEC (Figure 5). Therefore, I find the results presented not convincing enough to justify the authors' claims and conclusions. Moreover, some conclusions are not supported by the data.

In general, I find this work largely descriptive. It is hard to grasp the broader significance of the findings presented here when murine ES cells are the only model used throughout the manuscript, and no in vivo assays are employed. The only in vivo reference is mapping of scRNA-Seq data to an existing dataset, which provides little information other than the apparent lack of an interesting phenotype for Six4 and Zbtb7b KO, which are very similar to the WT control (Supp Figure 4). Specific comments below.

Major points.

1. Not enough methodological details are provided to adequately describe the techniques and methodologies. Very little information is provided in figure legends or in materials and methods. For example, a “high-dimensional flow cytometry panel using antibodies recognising 23 surface markers” is established. Results were reported in Figure 1 and 4 in the form of UMAP plots only. What surface markers have been used? How were cells gated and how were different cell subsets defined (e.g. primitive and definitive subsets)?

Figure 1c: the legend reports an “UMAP of 122 cells... coloured according to surface marker expression” but it looks like an RNA velocity analysis approach was used here, which is missing in the legend.

2. line 138-139: “This confirmed that DP priMes indeed had less capacity to form haematopoietic and endothelial lineages compared to F_SP HEM” I do not agree with this conclusion. I think what this experiment shows is that the frequency of hematopoietic progenitors in the DP population is lower than in the SP population, but not necessarily that the progenitors have less capacity to form hemato-endothelial lineages. Can the authors quantify the frequency of hematopoietic progenitors in each subset, for example by a limiting dilution approach?

3. I am not sure about the definition of the vSMC subset. These cells express CD43, CD14, CD44, CD16/32 (Supplementary Figure 1), strongly suggesting a hematopoietic identity instead.

4. As mentioned by the authors, it is strange that no chromatin factors emerge in the CRISPR screen in the transition from DP priMes towards F_SP HEM. Is this due to experimental conditions of the assay or is there a biological reason for this? It would be interesting to know what is going on with the other transitions.

5. Validation of targeting for the KO clones should be provided. More specifically, it would be important to provide a complete characterization of the KO lines (where did the editing occurred? which gRNA were used? what kind of impact had on protein expression? etc.)

6. Proper quantification of the analyzed populations in Fig3 should be provided. For example, in Figure 3a-b: Are there differences in the % of P-SP cells in the cultures generated from the KOs, as it would appear by the representative data shown in the dot plots? And in the DN? These are not commented on. Can the authors perform hemato-endothelial culture of these subsets (along with DP priMes and F_SP HEM), followed by a simple FACS analysis of CD41 and VECad as it was done in Figure 1?
7. How could the authors analyze EB cultures of NPM1 KO (Fig3h) when no live cells were present? Can they show dot plots for this analysis?
8. Figure 3i: what does the scale represent exactly? It seems like there is no expression at all for Six4 and Zbtb7b. How do the authors explain this?
9. The authors focus on the role of the selected factors on F_SP HEM commitment, but the impact on other transitions is not taken into account. Considering the broad expression of some of these factors (Fig. 3) and their known involvement in hemato-endothelial specification (i.e. Smad1), it would be interesting to know/quantify their impact on the earlier transitions.
10. Figure 4: Why did the authors sort total Flk1+ cells and not F_SP only since they say this is the affected population?
11. line 261-262: "Six4 KO cells showed a marked differentiation block towards the endothelial lineage". How is blood generated in these KO cells? From the UMAP in Figure 4 it seems that primitive erythrocytes are generated as well as "preHSPC" (what are these cells? What markers do they express?) Most importantly, where is the hemogenic endothelium in these haemato-endothelial cultures? Supposedly with the 23-marker flow panel the authors could define a putative hemogenic endothelial population. Is this still detected in the KO cultures? In lines 402-404 the authors also state that "As endothelial cells are critically needed for blood formation it is tempting to speculate that Six4 is involved in the differentiation of haemogenic endothelial cells (HECs)". Have hemogenic endothelial cells been detected in Six4 KO cultures?
12. lines 269-272: "This result indicates that Zbtb7b KO mesoderm is already committed towards haematopoietic and endothelial fate and their enhanced capacity during haemangioblast culture could be a direct effect of the increased frequency of F_SP cells in the starting culture" How many cells were actually plated here? Given the different F_SP frequencies, can the authors repeat the experiment with equal numbers of F_SP cells at the start of the culture?
13. lines 316-318: "Interestingly, our differentiation experiments had suggested that Six4 KO cells were able to form endothelial but not haematopoietic lineages, even if the most advanced C6 was present (Fig. 4a)". Isn't this a contradiction (see previous point 11 and lines 261-262)? Figure 4a clearly shows that hematopoietic cells are in fact generated in Six4 KO.
14. The scRNA seq on Flk1+ cells presented in Figure 5 are puzzling.
 - If C5 is "characterised by the highest expression of Etv2 and additional TFs associated with haematopoietic development, including Gata3 and Lmo2" and "is clearly committed towards blood development" (lines 330-332), then why is it overrepresented in Ldb1 KO which show a complete block in haemato-endothelial differentiation? On the other hand-why in Zbtb7b KO C5 does not show a higher frequency than the WT?
 - Again on the Zbtb7b KO, these cells show an enrichment of C4 (pre-HE mesoderm) but not the more hematopoietic committed clusters C5 and C6. How do the authors explain this?
15. lines 356-357: "Zbtb7b-KO cells, which are largely composed of the same clusters as the Ctrl, formed the most advanced C6 (HEM 2) via a different trajectory" How was this determined? Has pseudotime analysis been performed?
16. lines 355-356: "For all analysed conditions, Zbtb7b KO mapped with the highest frequency on the in vivo trajectory (21.1% vs. 20.4% for Ctrl)" Can the authors explain what this statement means exactly? Do they consider this a significant difference?
17. I generally disagree with the authors' conclusions regarding Smad1 KO. The fewer live cells in Smad1 KO (Figure 4c) could be indicative of a more generic role of Smad1 than the one proposed here, i.e. a more profound effect on all mesoderm. Proper quantification and statistical tests should be reported for each cell type if they want to make points such as "Smad1 KO cells were able to form both haematopoietic and endothelial cells albeit at a reduced frequency and number" (lines 264-265)
18. Line 276, "Heterogeneity of mesodermal cells is regulated by core TF network". I am not sure what the authors mean with this. They are not really assessing any kind of network, but just the effect of independent mutations. The impact of those mutations on gene expression or accessibility of other TFs is not analyzed.

Minor comments

- line 98- typo: Ldb1 is the correct gene name.
- lines 150-152: "Finally, to further underline the relevance of our ESC model we mapped the in vitro-generated cells from both EB culture and haemangioblast culture onto a mouse gastrulation atlas of cells isolated from E6.5–E8.5 embryos".

What kind of data were mapped here?

Version 1:

Reviewer comments:

Reviewer #1

(Remarks to the Author)

The authors have made an earnest and effective attempt to address the critiques from the prior review and have added data that addresses many of the points raised. The addition of some discussion of the limitations of the study has been incorporated. I agree with the revisions based on the very legitimate criticism raised by reviewer #3 of the lack of true analysis of a “transcription factor network”.

Reviewer #2

(Remarks to the Author)

All my comments are adequately addressed.

Reviewer #3

(Remarks to the Author)

I have reviewed the revised version of the manuscript titled “Targeted CRISPR-Cas9 screening identifies core transcription factors controlling murine haemato-endothelial fate commitment” by Teske et al.

I appreciate the efforts made by the authors to respond to reviewer comments. Although a number of issues raised by this and other reviewers have been addressed to a certain extent by including additional data, information, or discussion, several concerns remain.

The main experimental addition to the revised manuscript was a new “long-term” (9.5 days) hematopoietic differentiation assay, showing that F_{SP} HEM cells have significantly higher blood-forming potential compared to DP priMes, as expected. They quantified CD45⁺ and CD71⁺ output, confirming the already known biological significance of the PDGFR α downregulation as a developmental transition. Note that the materials and methods section titled “Long-term haematopoietic assay in liquid media” appears twice (lines 622-634 and 685-692), with different content.

No in vivo validation was performed for any of the factors emerging from their screen. Although the authors have now acknowledged this limitation in the Discussion, I believe that they should have at least tried. As is, the validity and significance of the findings remains limited to an in vitro murine system.

I found the comment about phenotype rescue from one of the other Reviewers particularly relevant. In the absence of in vivo validation, the claim that the TFs identified are important regulators of mesodermal commitment would be indeed strengthened if they at least showed some sort of functional rescue. The authors tried to address the comment and attempted two lines of rescue: (i) generation of double knockouts (e.g., Ldb1/Zbtb7b), showing only a partial phenotypic – but not functional – rescue (no blood or even endothelial cells generated) and (ii) lentiviral ETV2 overexpression in various KO lines. Again, the phenotype observed was only an increase of VE-Cad⁺ cells and only in some knockouts. These results are very preliminary and were not included in the revised article. Thus, I feel that at present, the functional relevance of the identified factors as regulators of HEM specification is not completely supported by the data and this important concern has not been conclusively addressed.

Major comments and outstanding issues:

1. I appreciate that the authors have recognized that, as pointed out in my previous review, some methodology was completely missing from the manuscript (ie. Spectral flow cytometry); they have now added information in their revised manuscript. They have now added information regarding how they have defined their populations after HB culture by spectral flow (Revised Fig 1f) (lines 148-155). I find these definitions very arbitrary and questionable. For instance, why do “EMPs” do not express CD16/32 (see <https://pubmed.ncbi.nlm.nih.gov/26095363/>) and are largely positive for CD43 (stem/progenitor marker) and CD71? This is a phenotype more consistent with erythroid progenitor activity, if anything. It is also puzzling that cells annotated as primitive erythrocytes and macrophages express CD41 as stated in the text, though this does not seem to correspond to what shown in the bubble plot (only a fraction of these cells appears to be CD41⁺). It is also strange that primitive macrophages appear largely positive for CD43, a marker associated with late progenitor/preHSC activity in mouse (<https://pubmed.ncbi.nlm.nih.gov/25241746/>). In general, the annotated populations seem very heterogeneous.

2. Relative to my previous comment no.3, I feel that my concern (quantification of data in Fig1) has not been properly addressed. Moreover, no quantification has been reported for the endothelial potential, thus not supporting the statement “The HB culture experiments confirmed that DP priMes indeed had less capacity to form haematopoietic and endothelial lineages compared to F_{SP} HEM, which generate a higher frequency of CD41⁺ and VE-cad⁺ cells and fewer VSM-enriched cells” (lines 140-143)

3. Relative to my previous comment no.4. While not central to the main message of the study, I disagree with the statement

included in the revised article that "there is no specific marker for VSM cells suitable for flow cytometry". What about aSMA, NG2, CD146, PDGFRb..? They might not be the more specific markers, but the inclusion of at least some of these markers would definitely help demonstrating the non-hemato-endothelial identity of the cells that they call vSMCs. Also, they say their panel is designed for hemato-endothelial cells, and therefore the identification of vSMC is based on the exclusion of lineage markers, but this is in contrast with the expression of hemato-endothelial markers on those cells (CD43, CD44, CD16/32, CD14...).

4. Relative to my previous comment no. 19. The manuscript still does not contain any TF network analysis. Although the authors agreed with this and removed most references to "TF networks," there are still instances in the manuscript where reference to transcriptional networks appear, e.g.: end of introduction (line 99); Title of section referring to Figure 4; Figure 5 title.

5. Relative to my previous comment 17. "For all analysed conditions, Zbtb7b KO mapped with the highest frequency on the in vivo trajectory (21.1% vs. 20.4% for Ctrl) indicating their enhanced capacity to commit towards the haemato-endothelial lineage (Supp. Fig. 4d)." As the sentence is descriptive and not biologically significant, and that difference it's really not relevant from an n=1 as confirmed by the authors in their response, the claim and this sentence should be removed.

Other comments:

Line 177: reference to fig. 1f appears to be wrong.

Line 202: reference to supp. Table 1 is wrong. Suppl. Table 2 appears to be missing .

Version 2:

Reviewer comments:

Reviewer #3

(Remarks to the Author)

I am satisfied with the authors' response to my comments, and have no additional concerns to report.

I would indeed find helpful to add REV2_Fig1 including manual gating strategy in the manuscript as supplemental data.

Lines 117-118 "we conducted gene expression analysis employing single cell RNA sequencing employing SORT-seq": I would recommend avoiding the "employing" repetition.

Response to reviewer Comments

We thank the reviewers for their constructive comments and valuable suggestions on our previous manuscript version. We have carefully addressed all points raised in the following point-by-point response. To support our revision, we provide additional experimental data, further analysis, and clarified conclusions throughout the revised manuscript.

In particular, based on the reviewers' recommendations, we have implemented the following key revisions:

- **New Experimental Data:** We employed a newly established long-term haematopoietic assay to assess the blood-forming potential and mature blood cell output of F_SP HEM versus DP priMes.
- **Extended CRISPR-Cas9 Analysis:** We expanded our CRISPR-Cas9 gene ranking analysis to include all mesodermal transitions. This dataset may serve as a valuable resource for the broader community investigating molecular regulators of early lineage commitment.
- **Hypothesis Testing:** We conducted additional experiments to rule out the possibility that the identified F_SP HEM driving transcription factors delay HEM maturation, thereby strengthening the functional interpretation of our findings.
- **Expanded Data Analysis and Clarifications:** We conducted deeper analysis of existing datasets, enhanced the Materials and Methods section for clarity and reproducibility, and incorporated textual clarifications throughout the manuscript.

To address these revisions, we have added updated figures/ figure panels to the revised manuscript:

- New panels in Figure 1: d, f
- New Supplementary Figure 2
- New panel in Supplementary Figure 3: a
- New panels in Supplementary Figure 4: b–d, f–k
- New Supplementary Figure 6
- New Supplementary Tables 1 and 2

The new additions and responses to the reviewers' comments can be found in the point-by-point response below.

Reviewer #1 (Remarks to the Author):

Overview: This manuscript reports a novel transcription factor network that drives the transition of murine primitive mesoderm to hemato-endothelial mesoderm (HEM)—extending previous work that has identified Runx1, Gata2 (and more recently, ETV2) as key regulators of early blood lineage commitment. Using a CRISPR-Cas9 knockout screen that targets TFs in a mouse ESC blood differentiation model, the authors identify four novel transcriptional regulators (Lbd1, Smad1, Six4, and Zbtb7b) that control lineage specification from primitive toward hemato-endothelial mesoderm. Teske and colleagues subsequently perform loss-of-function studies by knocking out each TF individually, which perturbs normal hematopoiesis in vitro, thus validating these genes as functional regulators of hemato-endothelial lineage commitment. Taken together, this technically rigorous screen resulting in identification of a novel TF network crucial for HE commitment advances our understanding of the complex networked regulators of murine hematopoiesis, which will inform ongoing and future efforts to generate hematopoietic cells from pluripotent stem cells (in particular, because generation of specific blood lineages hinges upon modulation of distinctly patterned subsets of mesoderm at the earliest stages of embryonic development).

Summary of study: Teske and colleagues first characterize and validate an in vitro EB-culture mESC blood

differentiation model, based off prior protocols by Gordon Keller's and Georges Lacaud's groups. They describe the progressive formation of distinct mesodermal entities defined by FLK1 and PDGFR α , where cells progress from populations that are (1) double-negative to (2) PDGFR α single-positive to (3) double-positive to (4) FLK1 single-positive, consistent with mesodermal commitment. RNA-seq reveals progressive downregulation of pluripotency markers and upregulation of mesodermal and hemato-endothelial genes along this developmental trajectory, while FLK1-single positive populations exhibit increased developmental potential for blood lineages (assayed for CD41 and VE-Cadherin).

Next, the authors perform a pooled CRISPR KO screen to identify putative drivers and repressors of hemato-endothelial (HEM) commitment and recover a list of candidate regulators of the transition from primitive mesoderm (double positive) toward FLK1-single positive HEM, identifying top depleted and enriched genes. The authors subsequently generate independent KO ES cell lines and differentiate them to assess relative competence to generate Flk1+ single-positive mesoderm, validating Smad1, Ldb1, Six4 as drivers—and Zbtb7b as a repressor—of the HE mesodermal commitment from primitive mesoderm. ETV2 was recovered as a top depleted hit and confirmed as a driver of primitive-to-HE mesoderm transition in accordance with previous studies, serving as a positive control.

To further uncover the effects of each gene KO on the downstream differentiation potential of HE mesoderm, the authors drive mesoderm cells toward hematopoietic and endothelial lineages, followed by analysis of fate specification, revealing that loss of function of each TF significantly biases the lineage differentiation of mesoderm subsets. Finally, the authors perform scRNA-seq characterization of FLK1+ cells in each KO line and identify 10 clusters corresponding to different developmental timepoints, finding that several clusters are absent in or specific to individual TF KOs, and that frequencies are distinct across genetic perturbations. Through pseudotime analysis, they construct a mesodermal developmental trajectory and demonstrate that each regulator TF distinctly modifies the trajectory of lineage commitment.

Major points:

1. The screen is performed on the basis of loss of PDGFR α expression in FLK1+ cells as demarcating the transition from primitive to hemato-endothelial mesoderm. While the authors do show in Figure 1c/d that Flk1+ single-positive cells are more competent to form VeCad+ CD41+ cells (and to give rise to hematopoietic and endothelial lineages), there are still significant numbers of cells that do not induce VeCad/CD41 expression at EB Day 5, suggesting that the Flk1 SP population is heterogeneous. The overall significance of the screen could be strengthened by molecular characterization of more mature blood cells differentiated from the FLK1+PDGFR α - (F_SP HEM) versus FLK1+PDGFR α + (DP priMes) fractions respectively, beyond merely showing that the SP fraction generates slightly higher percentages of VeCad+ and CD41+ cells. While the bulk and scRNA-seq data on EB Day 5 cells is convincing in demonstrating that early mesodermal markers and hematopoietic and endothelial markers are both segregated and enriched in DP_SP and F_SP HEM cells respectively, further functional and molecular characterization of specific blood cell types descending from these fractions of mesoderm (beyond solely flow cytometry analysis of surface markers in Fig 1e at 2.5 days post-EB culture) would reinforce the biological significance of PDGFR α downregulation. (For example, demonstration of enhanced emergence of HSCs or of other cell types, from the VeCad+CD41+ fraction arising from F_SP as compared to DP preMes (or even from the bulk F_SP HEM population as compared to DP priMes) would provide further assurance that PDGFR α downregulation does serve as a developmentally relevant transition.)

Author response: We thank the reviewer for raising the important point about the biological significance of FLK1/PDGFR α - expression during early blood development. Below, we address this concern in detail. Regarding the relevance of FLK1/PDGFR α - expression in mesodermal subsets, we acknowledge their biological significance and refer to several important studies demonstrating the usefulness of FLK1/PDGFR α - expression in distinguishing the developmental potential of mesodermal subpopulations (PMID: 21911838; PMID: 23335233; PMID: 9493833). The primary goal of our study was to identify factors **regulating HEM differentiation** and not on molecular regulators impacting **mature blood cells development**. That said, we

agree with the reviewer that it is a very important and interesting question whether mesodermal subpopulations differ in their potential in generating blood, which partially has been previously addressed in other studies (PMID: 21911838; PMID: 23335233; PMID: 9493833).

To address this concern in our study, we adapted a long-term haematopoietic assay in liquid culture (based on methods from PMID: 19182774; PMID: 19277585) supporting the maturation of haematopoietic progenitors and **compared F_SP HEM and DP priMes** cells in terms of (i) **their blood forming potential** and (ii) **their ability to generate mature blood lineages**. Defined numbers of FACS-sorted F_SP HEM or DP priMes cells (18k, 10k or 5k) from EB cultures were plated in gelatine-coated 96-well plates. A minimum of 24 wells were seeded for each condition. Cells were cultured first 2.5 days in haemangioblast culture media (to support the generation of haematopoietic progenitors) and subsequently in media supplemented with SCF, IL-3, G-CSF, IL-11, IL-6, TPO, M-CSF and erythropoietin to support the maturation of haematopoietic progenitors. After 9.5 days in culture, wells were scored for haematopoietic cells (REV Fig.1A, B). REV Fig.1. A shows representative microscopy images of a positive culture containing haematopoietic cells started from F_SP HEM and negative culture started from DP priMes where no haematopoietic cells formed at day 9.5. Whereas F_SP HEM generated haematopoietic cells under all tested conditions, DP priMes showed reduced blood-forming potential, particularly at lower cell inputs: For 5k cells: 100% of F_SP HEM formed colonies, but only 8% of DP priMes (REV Fig. 1B). Additionally for blood lineage analysis, we scored 8 randomly selected wells for each condition and analysed the number of live cells and the frequency of leukocytes (CD45+) and erythroid (CD71+) cells, the two main blood lineages forming in the assay (REV Fig. 1 C-F). Please see REV. Fig 1C for a representative flow cytometry analysis of CD45 and CD71 surface expression of a positive culture started from F_SP HEM (upper) and negative culture started from DP priMES (lower). F_SP HEM consistently showed higher number of live cells and increase frequencies of CD45+ and CD71+ cells in all analysed conditions compared to DP priMes (REV Fig. 1D-F). Collectively, these results demonstrate that mesodermal populations, defined by FLK1/PDGFR α expression, differ significantly in their blood-forming capacity. **F_SP HEM is functionally distinct from DP priMes with superior potential to form mature haematopoietic cells** due to increased potential to form haematopoietic progenitors. **This highlights the importance of FLK1/PDGFR α expression in marking functionally distinct mesodermal subpopulations during early haematopoietic development.** This data is now included in our manuscript as Supp. Fig 2.

REV Fig.1

2. In this study, characterization of in vitro differentiated hemato-endothelial lineages (HB culture) is performed at 2.5 days past FACS isolation of Day 5 EBs. The developmental significance of the authors'

findings could be significantly strengthened if they examined functional effects of modulating any of the four TFs on blood cells further differentiated down their developmental trajectory, which would lend credence to biological and functional implications of perturbing early mesoderm patterning for hematopoiesis. (For example, some functional effects of Zbtb7b KO on EB-derived cells after more than 2.5 days of HB culture would be interesting, and it would be especially of relevance to understand if knock-out of Zbtb7b might skew differentiation or enhance the production of certain more mature blood lineages.)

Author response: Is not disclosed publicly.

3. Moreover, because there remains incomplete understanding of how faithful the in vitro differentiation of murine ES cells mimics hematopoiesis in vivo, a limitation of the study is validation of the TFs in the context of murine hematopoiesis. While generating knockouts and evaluating hematopoiesis in murine embryos is beyond the scope of this study, some discussion of the limitations of the study for revealing in vivo mechanisms of hematopoiesis is warranted.

Author response: We thank the reviewer for raising this important point and agree with the suggestion. We include a new paragraph in the discussion of our revised manuscript addressing this limitation of the study as follows:

Lines 496-504:

Furthermore, our approach of modelling early haematopoiesis ex vivo employing murine ESCs enabled us to obtain sufficient cell numbers for large-scale genetic screenings with the optimal sgRNA representation of a delicate developmental time window that would have been challenging if not impossible in vivo. **Still, our study would be strengthened by in vivo characterisations and experiments addressing the function of the identified core TFs during early mesodermal and blood commitment in mouse development. For these in vivo experiments, factors that lead to early embryonic lethality like Etv2, Ldb1 and Smad1 would have to be contrasted to factors that have not been implicated in early haematopoiesis or have been associated with impaired survival, like Six4 and Zbtb7b.**

4. While not necessarily a pivotal issue with regard to the validity of the screen, it would strengthen the claim that each TF is specifically a negative or positive regulator of mesodermal commitment if authors could demonstrate that at least one of the KO phenotypes could be rescued by reintroducing the missing TFs exogenously through a lentiviral vector, etc.

Author response: We thank the reviewer for this suggestion. To answer if manipulating the expression of one of the identified TFs could rescue the observed KO phenotypes we performed different sets of experiments, described below:

1. We first addressed if the block of HEM differentiation upon knock-out of Etv2, Ldb1, Six4 and Smad1 could be rescued or ameliorated by additionally knocking-out Zbtb7b, as we identified that Zbtb7b knock out cells had an increased potential to develop towards F_{SP} HEM. We therefore generated Zbtb7b DKO ESCs in the individual single KO (SKO) lines by CRISPR-Cas9 targeting of Zbtb7b and generated ETV2/ Zbtb7b DKO, LDB1/ Zbtb7b DKO, Six4/ Zbtb7b DKO and Smad1/ Zbtb7b DKO lines, abbreviated as DKOs. We then performed EB differentiation experiments to monitor the frequency of F_{SP} HEM and analysed in total 3 different DKO clones per genotype (REV Fig. 3 A).

Interestingly, in the Ldb1/ Zbtb7b DKO cultures we detected a significant increase in frequency of F_{SP} HEM compared to LDB1 single KO cultures at EB day5 (REV Fig. 3A). The frequency of F_{SP} HEM in Ldb1/ Zbtb7b DKO cells was very similar to the WT level (REV Fig. 3A). The phenotypes of the remaining single KO lines could not be rescued by additionally knocking-out Zbtb7b.

We furthermore performed haemangioblast cultures of FLK1 + mesodermal cells isolated from WT, LDB1 KO and Ldb1/ Zbtb7b DKO EB cultures to test if we could obtain endothelial/haematopoietic cells and therefore rescue the developmental block of knocking out Ldb1. Despite the increased frequency of F_{SP} HEM in LDB1/ Zbtb7b DKOs cells, the cells could not form haematopoietic cells equally to the LDB1 single KO (REV Figure 3 B). In a follow-up, we plan to address (i) if an interaction of LDB1 and ZBTB7b exists and

(ii) the identity/ developmental potential of F_SP HEM cells developing in the LDB1/ Zbtb7b DKO background.

REV Fig. 3

2. As a second set of experiments we performed ETV2 overexpression experiments in the WT control and individual KO cell lines (Ldb1, Six4 and Smad1) and aimed to address if adding back ETV2 will rescue their differentiation block towards F_SP HEM. As the reviewer suggested, we used a lentiviral approach and cloned the coding sequence of ETV2 in a lentiviral construct under the PGK promoter containing a PGK_puromycin_T2A construct. This set-up enables a puromycin selection after lentiviral transduction, which ensured ETV2 expression of the transduced cells surviving the selection.

We next performed EB culture experiments after ETV2 transduction and selection. Unfortunately, ETV2 overexpression was not very well tolerated by our lines and we obtained significantly less live cells after EB 5 days regardless of the underlying genotype (REV Fig. 4A). In addition, we detected no decrease of F_SP cells for the analysed phenotypes. We still continued with this not ideal EB cultures and sorted total FIK1+ cells from both the untransduced controls and ETV2-add back cells lines and performed haemangioblast culture to analyse CD41 (haematopoietic marker) and VE-cad (endothelial marker) expression. We analysed the potential of the add-back lines to generated endothelial and haematopoietic cells. We detected no difference upon ETV2 overexpression in the Ctrl cells, but interestingly we detected partial rescues in all other analysed KO cell lines upon ETV2 overexpression. The rescue upon ETV2 overexpression on endothelial and/ or haematopoietic cell differentiation varied in the distinct KO lines. ETV2 KO cells upon ETV2 add-back were able to differentiate to haematopoietic and endothelial cells, albeit at a very reduced level. Ldb1 KO cells +Etv2 could only generate endothelial but no haematopoietic cells. Six4 KO+ ETV2 cells could reach Ctrl levels of haematopoietic cell differentiation and a marked increase of endothelial cells. Smad1 KO cells generated few haematopoietic cells but markedly more endothelial cells than wild type controls upon ETV2 overexpression. Collectively, despite some technical problems we could detect partial rescues in our KO cell lines. For the future we will now change our overexpression system and move to an inducible overexpression system and include additional promoters besides PKG.

Collectively, we hope that we could address some of the concerns of the reviewer and hope to optimise our overexpression set-up for future studies. Unfortunately, we realised that addressing potential rescue and add-back phenotype will require a substantial longer time frame.

REV Fig. 4

Minor points:

1. While the authors quantified sgRNAs in the T0 ESCs and were thus able to identify differences in guide abundance between DP and FLK1+ single-positive mesoderm, it would be informative and strengthen the robustness of this specific screening platform to show that sgRNA representation in the transduced mESCs (post-selection for Cas9-GFP+ cells) mirrors the representation of the EpiTF plasmid library, which would demonstrate that sgRNAs were introduced into ESCs without significant biases (e.g., a simple Lorenz curve in the supplement to show sgRNAs are uniformly expressed in ESCs without biases).

Author response: We thank the reviewer for raising this important point regarding sgRNA representation and apologise for not including this information in our initial manuscript.

To directly assess sgRNA representation post-transduction, we have added a new panel (now included as Supplementary Figure 3a/ REV Fig. 5) comparing the composition of the initial CRISPR plasmid library with the sgRNA abundance in FACS-sorted GFP+ ESCs at the start of differentiation (48 hours post-transduction). While the histogram for the T0 ESC sample shows a slightly broader distribution than the plasmid library, this variation does not indicate any major skew or dropout. The overall sgRNA representation remains comparable, with no significant biases detected after ESC transduction. Additionally, to further ensure robust representation, we performed all screens with a high coverage of approximately 500 x per guide thus minimizing the risk of losing low abundance sgRNAs and enhancing the reliability of downstream results.

REV Fig.5

2. Typo on page 9, line 234: "but was broadly expressed in all populations; while Ldb1, Six4."

Author response: The typo was corrected.

3. Typo on page 11, line 346: "sort due to a less stringent gating strategy."

Author response: The typo was corrected.

4. Page 11, line 342 "Supp Fig 5b" should instead be "Supp Figure 4b."

Author response: The typo was corrected.

Reviewer #2 (Remarks to the Author):

In this manuscript, Teske et al. describe the identification of novel transcription factors involved in specifying hematovascular mesoderm (HVM). Using CRISPR/Cas9 screening, authors revealed that *Smad1*, *Ldb1*, *Six4*, and *Zbtb7b* had a major impact on the formation of Flk1 single positive HVM. Results from CRISPR screening were confirmed using corresponding KO lines. Although significant research efforts have been devoted to studying factors involved in the formation of hemogenic endothelium and endothelial-to-hematopoietic transition (EHT), the initial steps in mesodermal specification remain understudied. Thus, the findings are novel and significant. I have only minor comments, mostly related to some data interpretation.

1. To assess the impact on commitment versus viability, the authors evaluated proliferative potential and viability after gene knockout. What about the possibility that certain genes might delay the maturation of the Flk1 SP population instead of abrogating it?

Author response: We thank the reviewer for raising this important point regarding a potential delayed appearance of F_SP HEM during EB cultures in the identified TF KOs.

The kinetics of FLK1 expression in embryoid body (EB) differentiation of ESCs were first established in the seminal work of Gordon Keller's group (PMID: 9169850). FLK1 expression exhibits a dynamic and narrow window, peaking around day 4–5 and declining rapidly thereafter. Haemato-endothelial progenitors are known to arise during this transient FLK1 high phase. Consistent with these findings, we observe the highest frequency of F_SP HEM at the FLK1 expression peak (day 5 in our system, please see Fig. 1a, b of the manuscript, REV Fig.6 A, B). This timing is critical, as delaying analysis risks missing this window of haemato-endothelial competence. To rule out the possibility of delayed F_SP HEM emergence in the KO lines, we extended EB differentiation to day 6 and repeated our analysis. Across all genotypes, total FLK1 and F_SP HEM frequencies were consistently lower on day 6 compared to day 5 (REV Fig. 6 A–C), suggesting that the differentiation potential had already declined. We additionally performed haemangioblast cultures of FLK1+

cells isolated from day6 (REV Fig. 6 D). No increase in haematopoietic or endothelial potential was detected compared to day 5 and we confirmed our KO line-specific observations: Etv2 KO and Ldb1 KO, showed a complete block in haemato-endothelial differentiation; Six4 KO and Smad1 KO, retained limited capacity to form haematopoietic and endothelial cells, though reduced compared to Ctrl cultures (REV Fig. 6D). Collectively, our data do not support a delayed emergence of HEM in any of the KO lines tested. Rather, the data indicate a disruption or block in differentiation, not a shift in timing. We included this new data in the revised manuscript (Supp. Fig 4 f-h).

REV Fig.6

2. There is some confusion regarding the phenotype used to identify the cell populations defined in the main and supplementary Fig. 1. The population appears to express very high levels of endothelial/early hematopoietic markers CD43, CD44, and CD202b. This could be due to issues with cell separation by flow or the presence of the CD31+CD144- endothelial population in cultures.

Author response: We thank the reviewer for the thoughtful comment and the opportunity to clarify our classification of VSM cells in Fig. 1e and the updated Fig. 1f (previously Supplementary Fig. 1f). We also refer to Reviewer 3 (Q4) that raised a similar concern.

Our high-dimensional FACS panel was primarily designed to distinguish haematopoietic and endothelial lineages. The identification of VSM cells, which represent an alternative differentiation fate in haemangioblast cultures (PMID: 11081514; PMID: 15677567; PMID: 12569129; PMID: 17084363), is based on exclusion of known lineage markers. Currently, there are no widely accepted surface markers that can uniquely define VSM cells by FACS. As such, we identified the VSM population indirectly, using negative gating -specifically, by excluding cells expressing endothelial and haematopoietic lineage markers. For any cluster to be bona fide haematopoietic a particular focus is on CD41 expression, the most specific and reliable marker for early

blood development (PMID: 12900455; PMID: 11934866). For any cluster to be assigned an endothelial identity (or cells undergoing EHT) a particular focus lies on VE-cad, the most specific and reliable marker for endothelial cells (PMID: 8555485). The VSM cluster is VE-cad neg/ CD41 neg, supporting a non-haematopoietic, non-endothelial identity. Further evidence comes from the analysis of haemangioblast cultures from EVT2 KO cells, which also differentiate into cells of the VSM cluster (Fig. 4 a, b). ETV2 is the Master TF, needed for endothelial and haematopoietic differentiation. We therefore do not consider a haematopoietic/ endothelial identity of the VSM cell cluster at the moment.

We recognize that the VSM cluster expresses some markers commonly associated with haematopoietic cells, including CD44, CD202b (Tie2), CD43, CD14, and CD16/32. However, expression of these markers is not exclusive to the haematopoietic lineage: CD44 has been reported to be expressed on VSM cells (PMID: 8609213). Additionally, CD202b (Tie2) expression was detected in embryonic VSM precursors and primitive mesoderm cells, prior to fate commitment (PMID: 16926294, PMID: 22509029). Low or ectopic expression of CD43 has been reported in non-haematopoietic cells, typically in the context of tumour cells (PMID: 28807337, PMID: 12499775). CD14 and CD16/32 expression was reported in epithelial cells in specific contexts (PMID: 11349042, PMID: 25904149).

We agree with the reviewers that due to the absence of a unique and positive VSM surface marker, we cannot definitively confirm the identity of all cells within this population as true VSM cells. To more accurately reflect this uncertainty, we have updated our terminology throughout the manuscript. The population previously referred to as "VSM" is now labeled "VSM-enriched", acknowledging the inferred nature of this identity. This is an approach also used in other publications (PMID: 36217016).

3. C8 and C7 may represent different parts (anterior-posterior) of the primitive streak based on the high expression of MIXL1, EOMES, MESP1, LHX1, and TLL1 genes found in the primitive streak. It could be useful to check/show additional primitive streak markers, Meox, GATA6, SOX17 and FOXA2, to see if these populations reflect different parts of the primitive streak. C2 and C3 most likely represent a lateral plate mesoderm since cells express very low levels of T and high levels of the lateral plate mesoderm markers FOXF1, HAND1, and BMP4.

Author response: We thank the reviewer for the thoughtful comment and included the suggested markers in our scRNAseq analysis (REV Fig.7). For Gata6, we detected a higher expression in C8 compared to C7, which indeed could be an indication of different primitive streak parts. We did not detect any cluster-specific expression for Meox1/2, Sox18 and FoxA2 and did not extend the data we are showing in the manuscript (Fig.5), but are grateful for the reviewer's suggestion.

REV Fig. 7

4. I would discourage using the term "mature endothelial cells" to define C9 cluster cells. These emerging endothelial cells co-express KDR and likely lack arterial and venous markers, thus representing early endothelial progenitors or emerging endothelial cells rather than mature endothelial cells. They also express high levels of the PLXND1 gene, which is involved in cardiovascular development, and could be endothelial cells developing from the anterior primitive streak committed to cardiac fate.

Author response: We agree with the reviewer and thank for his insightful input. We changed our nomenclature of C9 to early endothelial progenitors.

5. Table 1 is in supplemental materials and, therefore, should be labeled as Table S1. Otherwise, it is challenging to locate this Table.

Author response: We agree with the reviewer and changed Table 1 to Supp. Table 3.

Reviewer #3 (Remarks to the Author):

In this manuscript, the authors perform a CRISPR-Cas9 KO screen in a murine embryonic stem cell (ES) system to identify novel regulators of haemato-endothelial mesoderm differentiation. Upon identifying potentially interesting candidates in their screen, they apply selection criteria to restrict the number of hits down to 4 genes and proceed to generate stable KO ES cell lines. They next combine spectral and classic flow cytometry with single cell RNA sequencing to characterize the effect of genetic deletion of the hit genes during haemato-endothelial differentiation. The methodologies in this study are generally solid and the experiments appear to be well designed and executed (although lacking in reporting detail – see below). However, I am not convinced that the data reported in this manuscript can address the main questions that the authors would like to answer. Quoting from the introduction, lines 88-90: "The major open question in the field today is how these - and possibly other yet unidentified - regulators cooperate to allow the initial transition towards the blood lineage by the differentiation of priMes to the HEM". Although some transcription factors potentially involved in haemato-mesoderm formation are identified here, there is no data about how these genes cooperate with existing factors, as only single KO lines are analyzed; no molecular studies are performed to evaluate how the identified TFs interact. Furthermore, I find the title of the MS "Targeted CRISPR-Cas9 screening identifies transcription factor network controlling murine haemato-endothelial fate commitment" misleading as in this manuscript no analysis of transcription factor networks is provided.

Of the four validated genes, two were previously known as regulators of hemato-endothelial differentiation (Ldb1 and Smad1, refs. 35 and 36 and others). The other gene validated as a putative positive regulator, Six4, was not previously implied in HEC differentiation, but it appears not essential for blood generation in their system (See Fig. 3b, 4a) though its KO results in a decrease in the frequency of Flk1+ mesoderm. At the same time, the KO of Zbtb7b, putative repressor of hemato-endothelial differentiation, only results in an increase of primitive erythrocytes (Figure 4), a lineage not necessarily derived by HEC, and does not result in an increase of Runx1+ HEC (Figure 5). Therefore, I find the results presented not convincing enough to justify the authors' claims and conclusions. Moreover, some conclusions are not supported by the data.

In general, I find this work largely descriptive. It is hard to grasp the broader significance of the findings presented here when murine ES cells are the only model used throughout the manuscript, and no in vivo assays are employed. The only in vivo reference is mapping of scRNA-Seq data to an existing dataset, which provides little information other than the apparent lack of an interesting phenotype for Six4 and Zbtb7b KO, which are very similar to the WT control (Supp Figure 4). Specific comments below.

Major points:

Q1: 1. Not enough methodological details are provided to adequately describe the techniques and methodologies. Very little information is provided in figure legends or in materials and methods. For example, a "high-dimensional flow cytometry panel using antibodies recognising 23 surface markers" is established. Results were reported in Figure 1 and 4 in the form of UMAP plots only. What surface markers have been

used? How were cells gated and how were different cell subsets defined (e.g. primitive and definitive subsets)?

Authors response: We thank the reviewer for his comment and the opportunity to clarify our methodologies used in the paper. Regarding our spectral flow cytometry analysis, we agree with the reviewer that additional clarification on the antibody panel, gating strategy, and subset definition is important and was missing in our previous manuscript. Below we provide further details:

We used a 23-marker antibody panel (listed in Supp. Table 1), which targets key lineage proteins to comprehensively capture haematopoietic populations and endothelial cells. After staining, we excluded doublets (based on side scatter area vs. height), non-viable cells (Zombie NIR+ events) using FlowJo (v10) to ensure that only viable cells were analysed. **Unlike conventional cytometry approaches that rely on sequential manual gating, we employed a data-driven workflow implemented in the CATALYST framework in R (v4.1.2).** Fluorescence intensities were transformed using a hyperbolic arcsine (arcsinh) function to manage the large dynamic range in the dataset. A subsequent percentile normalization step aligned marker expression values to a 0–1 scale across all samples. We further used Uniform Manifold Approximation and Projection (UMAP) to visualize high-dimensional data in a two-dimensional space without bias from predefined gating. FlowSOM clustering then grouped cells based on their expression of all 23 markers. This unbiased method enabled us to identify distinct populations that might otherwise be missed using only manual gating thresholds. Each cluster's identity was determined by inspecting the median expression level of all 23 markers. Clusters with highly similar marker profiles were merged, as appropriate, to yield well-defined subsets. Subsets corresponding to primitive and definitive hematopoietic populations were annotated based on known lineage marker expression patterns. To illustrate these definitions, we provide a dot plot (REV Fig. 8/ Fig. 1f) showing the median expression of each marker across all clusters. We moved the dot plot from previously Supp. Fig 1f to main Figure 1f, to make it easier to access this important information. Additionally, we included in the text the main defining lineage markers for each subset.

The new text reads now as follows (lines 148-155).

F_SP HEM was more proficient at forming haematopoietic and endothelial lineages, including primitive (e.g. primitive erythrocytes ($CD41^{+}ckit^{+}CD71^{+}$), primitive macrophages ($CD41^{+}ckit^{Dim}CD11b/c^{+}CD44^{+}CD45^{+}Cx3Cr1^{+}$) and definitive blood subsets comprised of EMPs ($CD41^{+}ckit^{+}CD71^{+}CD43^{+}$), pre-haematopoietic stem and progenitor cells (pre-HSPC, $CD41^{+}VE-cad^{+}Tie2^{+}CD34^{+}ckit^{+}$) and endothelial type 1 ($Tie2^{+}FLK1^{+}CD31^{+}VE-cad^{+}c-Kit^{-}$) and type 2 ($Tie2^{+}FLK1^{+}CD31^{+}VE-cad^{+}c-Kit^{+}$) clusters, whereas DP priMes had a higher potential to form **VSM-enriched cells** ($CD41^{+}VE-cad^{+}Epcam^{-}$)(Fig.1e, Fig.1f).

REV Fig.8 / Fig. 1f

We furthermore included the following information in the Material and Method section, to add substantial more information on our high-dimensional FACS analysis and apologise that part of the information was missing in our previous manuscript (lines 650-683):

Spectral Flow Cytometry: Cells were first labelled with Zombie NIR fixable viability dye (BioLegend, 1:500) for the exclusion of dead cells. Before antibody staining, cells were incubated with TrueStain FcX™ and TrueStain Monocyte Blocker (BioLegend) to reduce nonspecific binding. Surface antigens were labelled using an antibody panel consisting of the following 23-markers (Supp. Table 1): From BD biosciences: anti-CD117(cKit) (BUV395, Clone 2B8), anti-CD43 (BUV496, Clone 1B11), anti-CD45 (BUV563, Clone 30-F11), anti-CD9 (BUV661, Clone KMC8), anti-CD44 (BUV737, Clone IM7), anti-CD31 (BUV805, Clone 390), anti-Epcam (CD326) (BV480, Clone G8.8), anti-CD93 (BV605, Clone AA4.1), anti-CD41 (BV650, Clone MWRReg30), anti-CD14 (FITC, Clone rmC5-3), anti-CD71 (RB780, Clone C2), and anti-CD47 (PE-CF594, Clone miap301). From BioLegend: anti-CD64 (BV421, Clone X54-5/7.1), anti-CD16/32 (BV711, Clone 93), anti-CX3CR1 (BV785, Clone SA011F11), anti-CD202b (Tie2) (PE, Clone TEK4), anti-CD34 (PE-Cy5, Clone MEC14.7), anti-CD309 (FLK1) (PE-Cy7, Clone Avas12), anti-Ly6C (Alexa Fluor 700, Clone HK1.4), anti-Sca-1 (APC-Fire 750, Clone D7), and anti-CD11b (APC-Fire 810, Clone M1/70). From ThermoFisher: anti-CD11c (PE-Cy5.5, Clone N418), and anti-VE-Cadherin (CD144) (eFluor660, Clone BV13). Surface staining took place on ice for 20 minutes in phosphate-buffered saline (PBS), followed by three PBS washes. Cells were subsequently acquired using a Cytex Aurora 5L spectral flow cytometer (Cytex Biosciences). **Data Preprocessing and Export:** Doublets were excluded based on side scatter area versus height, and non-viable cells identified by Zombie NIR™ positivity were removed using FlowJo (v10, TreeStar). The resulting compensated, pre-gated dataset was exported in FCS format for further analysis with R software (version 4.1.2). **Transformation and Normalization of Data:** The flowCore R package was utilized to import the FCS files via the read.flowSet() function. Fluorescence intensity values were transformed using an arcsinh function to address the wide dynamic range. To facilitate direct comparisons across samples, data underwent percentile normalization, scaling expression values to a consistent 0–1 range. **Dimensional Reduction and Clustering:** All 23 markers, including lineage- and progenitor-associated proteins, were subjected to dimensional reduction through uniform manifold approximation and projection (UMAP), implemented via the umap R package. For unbiased identification of cellular subsets, FlowSOM clustering was conducted using the same transformed and normalized data. FlowSOM-derived clusters were grouped through metaclustering, with subsequent refinement by manual annotation guided by expert assessment. Cluster identities were determined based on median marker expression levels, merging clusters exhibiting similar marker profiles. This approach defined clear subsets of primitive and definitive haematopoietic cells, endothelia cells. A dot plot depicting median marker expression per cluster is shown in Figure 1f to illustrate cluster characterization.

Q2: 1. Figure 1c: the legend reports an "UMAP of 122 cells... coloured according to surface marker expression" but it looks like an RNA velocity analysis approach was used here, which is missing in the legend.

Author response: We agree with the reviewer and adapted the figure legend accordingly (lines 527-530):

Fig1 c UMAP of transcriptomic profiles of 1,221 cells isolated by FACS from EB cultures at day 5. Cells are coloured by EB population, as defined by surface marker expression. **Arrows represent RNA velocity vectors, indicating predicted developmental trajectories between populations.**

The RNA velocity analysis is included in Material and Method section (SORT-seq) as follows:
Lines 770-776:

For RNA velocity analysis, the Seurat data were exported and an annotationData object was manually constructed in Python. Subsequently, scvelo v0.2.5 (ref61) was applied to it for comprehensive RNA-velocity analysis. The latent time of each cell from the dynamical model was used to calculate an average per cluster, thereby arranging the clusters along the developmental trajectory.

Q3: 2. line 138-139: "This confirmed that DP priMes indeed had less capacity to form haematopoietic and endothelial lineages compared to F_SP HEM" I do not agree with this conclusion. I think what this experiment shows is that the frequency of hematopoietic progenitors in the DP population is lower than in the SP

population, but not necessarily that the progenitors have less capacity to form hemato-endothelial lineages. Can the authors quantify the frequency of hematopoietic progenitors in each subset, for example by a limiting dilution approach?

Author response: We thank the reviewer for raising this important point and also refer to point 1 of Reviewer 1 (page 2 of P-b-P reply) that had a similar concern. We fully agree with the reviewer that DP priMes forms less haematopoietic progenitors compared to F_SP HEM. We adapted a long-term haematopoietic assay in liquid culture (based on methods from PMID: 19182774; PMID: 19277585) supporting the maturation of haematopoietic progenitors and **compared varying numbers of F_SP HEM and DP priMes** cells in terms of (i) **their blood forming potential** and (ii) **their ability to generate mature blood lineages**. We kindly refer to REV Fig. 1 which is now included in our manuscript as Supp. Fig. 2. Our results demonstrate that mesodermal populations, defined by FLK1/PDGFR α expression, differ significantly in their blood-forming capacity. F_SP HEM is functionally distinct from DP priMes with superior potential to form mature haematopoietic cells due to increased potential to form haematopoietic progenitors.

Q4: 3. I am not sure about the definition of the vSMC subset. These cells express CD43, CD14, CD44, CD16/32 (Supplementary Figure 1), strongly suggesting a hematopoietic identity instead.

Author response: We thank the reviewer for raising the important point about the definition of the VSM cell subset and refer to our response to Reviewer 2, 2nd point (page 8 of P-b-P reply), that raised a similar concern. We agree with the reviewers that due to the absence of a unique and positive VSM surface marker, we cannot definitively confirm the identity of all cells within this population as true VSM cells. To more accurately reflect this uncertainty, we have updated our terminology throughout the manuscript. The population previously referred to as "VSM" is now labeled "VSM-enriched", acknowledging the inferred nature of this identity. This is an approach also used in other publications (PMID: 36217016).

Q5: 4. As mentioned by the authors, it is strange that no chromatin factors emerge in the CRISPR screen in the transition from DP priMes towards F_SP HEM. Is this due to experimental conditions of the assay or is there a biological reason for this? It would be interesting to know what is going on with the other transitions.

Author response: We thank the reviewer for this insightful comment. In response, we have now included Supp. Table 2, which shows the overlaps of our CRISPR-screen of the Top 200 depleted/enriched genes across all transitions during mesodermal formation during EB cultures:

- CRISPR-Cas9 plasmid library \rightarrow DN
- DN \rightarrow P_SP
- P_SP \rightarrow DP priMes
- DP priMes \rightarrow F_SP HEM (main focus of the current manuscript)

The reviewer correctly pointed out that we did not validate any chromatin regulator specifically impacting the transition of DP priMes \rightarrow F_SP HEM. However, our data indicate that chromatin regulators may play more prominent roles at earlier stages of the differentiation trajectory. These regulators were consistently found among the top depleted hits in transitions preceding the primitive-to-haemato-endothelial mesoderm (HEM) stage, some with known functions in exit of pluripotency and/or lineage differentiation of ESCs. For instance, in the comparison between the CRISPR-Cas9 plasmid library and DN cells, we highlight the following: *Ddb1*, *Kat8*, *H2afz* (H2A.Z), and *Prmt5* – all of which have recognized roles in exit of pluripotency and early lineage specification.

In the DN \rightarrow P_SP transition, we highlight additional chromatin regulators among the top hits included: *Arid3b*, *Brd8*, *Jarid2*, *Kdm6a* (UTX), *Kmt2b/d* (MLL4/2), and *Zfp281* (transcriptional regulator with known chromatin-modifying activity).

We further extended the discussion of our revised manuscript as follows:

Lines 420-427:

Therefore, the absence of chromatin regulators in the later transition of DP priMes → F_SP HEM may be attributable to our experimental design, where gene knockouts were induced at the pluripotent stage (day 0). This setup may not capture the effects of genes that act during both earlier and later stages. To better test the involvement of chromatin regulators in the DP priMes → F_SP HEM transition, an inducible screening approach could be employed, in which gene knockouts are activated at later stages of the protocol (e.g. around day 3). This would allow to: (i) more precisely test the stage-specific roles of chromatin regulators, and (ii) evaluate their contribution to the transition toward haemato-endothelial mesoderm.

In summary, while we did not validate chromatin regulators in the final transition toward HEM in the current screen, the proposed adjusted set-ups could test if their contribution is temporally restricted to earlier differentiation windows — an important consideration for the design of future functional screens.

Q6: 5. Validation of targeting for the KO clones should be provided. More specifically, it would be important to provide a complete characterization of the KO lines (where did the editing occurred? which gRNA were used? what kind of impact had on protein expression? etc.)

Author response: We thanks the reviewer for the possibility to clarify how we generated the KO clones. We used a two sgRNA targeting strategy and a puromycin reporter construct to enhance KO efficiency. The targeting sgRNAs were listed in Supp. Table 2 and we updated this list to give more information (cut site and size of deletion) and include it as new Supp. Table 4. Single KO lines harbouring the deletion were selected after genotyping and were validated by Sanger sequencing. We additionally performed western plot analysis of the validated candidates and included it in the paper ((REV Fig. 9, Supp. Fig. 6). WB detection was performed using nuclear extracts of EB cultures day 5 of indicated genotypes. Asterisks denotes unspecific bands and the approximate sizes are indicated in kDa (REV Fig.9).

REV Fig. 9

Q7A: 6. Proper quantification of the analyzed populations in Fig3 should be provided. For example, in Figure 3a-b: Are there differences in the % of P-SP cells in the cultures generated from the KOs, as it would appear by the representative data shown in the dot plots? And in the DN? These are not commented on.

Author response: We thanks the reviewer for raising this point and provide the data for the frequencies of DP priMes, P_SP and DN cells for both the depleted (REV Fig. 10 a-c) and enriched candidates REV Fig. 10 d-g). We furthermore added this new data in the revised manuscript (Supp. Fig 4b-d, i-k) and adapted the text as follows:

lines 229-232:

Of the seven depleted candidates, four resulted in a significant decrease in the frequency of F_SP: Smad1 (rank 1), Etv2 (rank 2), Ldb1 (rank 6) and Six4 (rank 9) (Fig. 3a, b). **The KO lines exhibited an accumulation**

of cells at earlier developmental transitions: Etv2 KO, Smad1 KO and Ldb1 KO had a significant increased frequency of DP priMes and Six4 KO of DN cells (Supp. Fig. 4 b-d).

lines 251-253:

We next considered the enriched genes, analysing four hits and validating one: Zbtb7b (rank2). EB cultures of Zbtb7b KO cells generated F_SP HEM at a significantly higher frequency than controls, leading to an increased ratio of F_SP HEM/ total FLK+ cells (Fig. 3e-g). Zbtb7b KO cells showed decreased frequencies of earlier transitions including DP priMes and P_SP (Supp. Fig 4i-k).

REV Fig. 10

Q7B: Can the authors perform hemato-endothelial culture of these subsets (along with DP priMes and F_SP HEM), followed by a simple FACS analysis of CD41 and VE-cad as it was done in Figure 1?

Author response: We agree with the reviewer that adding the data addressing the haemato-endothelial potential of DN and P_SP cells should be included in the manuscript. We therefore adapted the Fig.1 of our revised manuscript and included a new panel (REV Fig. 11/ Fig. 1d). In summary, and as expected, DN and P_SP cells have a very limited capacity to form haematopoietic and endothelial lineages. We also adapted the text as follows:

Lines 143-145:

Accordingly, and as expected, DN and P_SP had a further reduced capacity to form haematopoietic and endothelial lineages compared to DP priMes/ F_SP HEM (Fig. 1d).

REV Fig. 11

Q8: 7. How could the authors analyze EB cultures of NPM1 KO (Fig3h) when no live cells were present? Can they show dot plots for this analysis?

Author response: EB cultures of NPM1 KO cells resulted in significant less live cells at day5 compared to the Ctrl, but we still obtained enough cells to perform our FACS analysis (REV Fig. 12). More specifically, whereas each EB culture of the Ctrl cell line resulted in an average of $14,15 \times 10^6$ live cells per culture, NPM1 KO cultures had in average 13'960 live cells (~ 1000-fold decrease). This was consistent for all three analysed NPM1 KO clones.

REV Fig. 12

Q9: 8. Figure 3i: what does the scale represent exactly? It seems like there is no expression at all for Six4 and Zbtb7b. How do the authors explain this?

Author response: We thank the reviewer for this observation. UMAP feature plots in Fig.3i showing SCTransform-normalized expression levels (Pearson residuals) of validated candidate genes in mesodermal populations, as measured by scRNA-seq. Dark purple denotes low expression, yellow denotes high expression.

We agree with the reviewer that Six4 and Zbtb7b have low expression levels in EB cultures d5. Despite being functionally relevant, it is not uncommon for TFs to exhibit low transcript abundance, particularly in early developmental contexts. This is a known limitation in scRNA-seq analyses, as low expression levels may result in dropout events or poor detectability, despite biological importance.

Q10: 9. The authors focus on the role of the selected factors on F_SP HEM commitment, but the impact on other transitions is not taken into account. Considering the broad expression of some of these factors (Fig. 3) and their known involvement in hemato-endothelial specification (i.e. Smad1), it would be interesting to know/quantify their impact on the earlier transitions.

Author response: We kindly refer to our answer to Q.7A (page 15 of P-b-P reply) and REV Fig. 10, now included in Supp. Fig. 4)

Q11: 10. Figure 4: Why did the authors sort total Flk1+ cells and not F_SP only since they say this is the affected population?

Author response: We appreciate the reviewer's comment. Figure 4 assesses the developmental potential of mesodermal cells following candidate gene knockout (KO). We chose to perform this assay using total FLK1⁺ mesodermal cells rather than isolating only F_SP HEM cells for two main reasons. First, our aim was to evaluate the full developmental potential of FLK1⁺ mesodermal committed cells, which include both F_SP HEM and DP priMes populations. The use of FLK1⁺ cells in haemangioblast cultures is well established and widely accepted in the field (e.g. PMID: 19182774; PMID: 33420489). Importantly, this approach ensures comprehensive detection of endothelial and haematopoietic potential within mesodermal progenitors, independent of PDGFR α expression status. And second, the knockout of transcription factors driving HEM specification led to a marked reduction in F_SP HEM frequency. Therefore, isolating sufficient numbers of F_SP HEM cells for functional assays would have required prolonged cell sorting times, reducing experimental feasibility and reproducibility. By using total FLK1⁺ cells, we ensured adequate cell numbers, enabling us to perform haemangioblast cultures under consistent conditions—including 6 genotypes—in parallel, minimizing batch effects and enhancing comparability.

Collectively, we believe this setup provides a robust and biologically relevant assessment of mesodermal potential upon candidate gene knockout.

Q12: 11. line 261-262: "Six4 KO cells showed a marked differentiation block towards the endothelial lineage". How is blood generated in these KO cells? From the UMAP in Figure 4 it seems that primitive erythrocytes are generated as well as "preHSPC" (what are these cells? What markers do they express?) Most importantly, where is the hemogenic endothelium in these haemato-endothelial cultures? Supposedly with the 23-marker flow panel the authors could define a putative hemogenic endothelial population. Is this still detected in the KO cultures? In lines 402-404 the authors also state that "As endothelial cells are critically needed for blood formation it is tempting to speculate that Six4 is involved in the differentiation of haemogenic endothelial cells (HECs)". Have hemogenic endothelial cells been detected in Six4 KO cultures?

Author response: We thank the reviewer for the opportunity to clarify how blood cell generation occurs in the haemangioblast (HB) culture system, which is a widely used assay to study mesodermal differentiation.

In our assay, mesodermal cells sorted from embryoid bodies (EBs) are plated on gelatinised plates in the presence of VEGF and IL-6. Two distinct types of blood cells can arise:

- Primitive blood cells emerge directly from mesodermal progenitors. These cells are CD41⁺ but c-Kit⁻.
- Definitive blood cells originate from haemogenic endothelial cells (HECs) that form during culture. These cells adhere to the plate and undergo endothelial-to-haematopoietic transition (EHT), releasing round, c-Kit⁺CD41⁺ blood cells into the media.

During EHT, pre-HSPCs transiently co-express endothelial (e.g., VE-cadherin) and haematopoietic (e.g., CD41) markers. This dynamic process can be directly visualized as blood cells budding from adherent VE-cad⁺ cells. We refer the reviewer to seminal studies by the Lacaud and Schroeder groups for detailed descriptions of this process (PMID: 19182774; PMID: 19212410).

Despite extensive efforts by several groups (e.g. PMID: 31996681) there is currently no specific surface marker that reliably discriminates haemogenic from non-haemogenic endothelial cells. However, researchers commonly use combinatorial marker panels and functional criteria to define and isolate HECs. Our 23-marker flow cytometry panel also does not allow definitive identification of HECs. However, the presence of definitive blood cells and pre-HSPC (CD41⁺c-Kit⁺ cells) in the culture serves as indirect but functional evidence that HECs were present and active in the assay.

We were intrigued by the behaviour of Six4 KO cells in HB cultures: These KO cells retained the capacity to generate both primitive and definitive blood lineages, indicating preserved haematopoietic potential. However, we detected a specific reduction in endothelial type I cells (Tie2⁺FLK1⁺CD31⁺VE-Cad⁺c-Kit⁻) in the KO condition. In contrast, type II endothelial cells (Tie2⁺FLK1⁺CD31⁺VE-Cad⁺c-Kit⁺)—which are potentially enriched of HECs—were present at levels comparable to controls.

This suggests that Six4 might play a role in specifying non-haemogenic endothelial fate, and its absence may bias mesodermal cells toward a haemogenic trajectory. We propose that Six4 expression could potentially serve as a negative marker of haemogenicity, and we are interested in exploring this hypothesis in future studies.

Q13: 12. lines 269-272: "This result indicates that Zbtb7b KO mesoderm is already committed towards haematopoietic and endothelial fate and their enhanced capacity during haemangioblast culture could be a direct effect of the increased frequency of F_SP cells in the starting culture" How many cells were actually plated here? Given the different F_SP frequencies, can the authors repeat the experiment with equal numbers of F_SP cells at the start of the culture?

Author response: For our experiments involving haemangioblast cultures, the same amount of cells are plated for each genotype. We typically plate 600k cells per well in a 6-well plate containing gelatine and media containing IL-6 and VEGF. As the reviewer suggested we repeated haemangioblast cultures F_SP HEM isolated from Ctrl and Zbtb7b KO EB cultures day 5 (REV Fig. 13). We have no indication that F_SP HEM cells from Zbtb7b KO cells are more prone to generate haematopoietic/ endothelial cells as comparable numbers of live cells are detected at HB culture at d2.5 and similar frequencies of CD41⁺, VE-cad⁺ cells. We therefore think, as stated above, that the enhanced capacity of Zbtb7b KO cells during haemangioblast culture is a direct effect of increased frequency of F_SP HEM.

REV Fig. 13

Q14: 13. lines 316-318: "Interestingly, our differentiation experiments had suggested that Six4 KO cells were able to form endothelial but not haematopoietic lineages, even if the most advanced C6 was present (Fig. 4a)". Isn't this a contradiction (see previous point 11 and lines 261-262)? Figure 4a clearly shows that hematopoietic cells are in fact generated in Six4 KO.

Author response: We thank the reviewer for bringing this point to our attention. We agree that the original statement was incorrect and misleading. We have revised the manuscript accordingly to clarify this point. The updated section reads now:

Lines 348-352:

In Six4 KO cell cultures, all clusters were present that were in the Ctrl samples, albeit with altered frequencies, but - uniquely among the F_SP-driving TFs - Six4 KO were able to form cells belonging to

the most advanced C6, albeit at highly reduced frequency (Fig. 5a, d, e). This is in agreement with our differentiation experiment, where Six4 KO cells were able to form all haematopoietic lineages (Fig. 4a).

Q15: 14. The scRNA seq on Flk1+ cells presented in Figure 5 are puzzling.

- If C5 is "characterised by the highest expression of Etv2 and additional TFs associated with haematopoietic development, including Gata3 and Lmo2" and "is clearly committed towards blood development" (lines 330-332), then why is it overrepresented in Ldb1 KO which show a complete block in haemato-endothelial differentiation? On the other hand-why in Zbtb7b KO C5 does not show a higher frequency than the WT?
 - Again on the Zbtb7b KO, these cells show an enrichment of C4 (pre-HE mesoderm) but not the more hematopoietic committed clusters C5 and C6. How do the authors explain this?

Author response: We thanks the reviewer for the comments regarding our scRNAseq analysis of FLK1+ mesodermal cells. Fig.4 d/e in our manuscript addresses the proportions of cells belonging to each annotated cluster per genotype. **Ldb1 KO cells accumulate in C5 (HEM 1)** (32.59% for Ldb1 KO vs. 5.3% in Ctrl) due to a developmental block, **whereas the Ctrl cells develop further along the mesodermal trajectory and differentiate to the most advanced cluster C6 (HEM2)** (35.63% in Ctrl vs. 0.76% in Ldb1KO). As we did not detect any haematopoietic or endothelial potential in Ldb1 KO cells, we speculate that Ldb1 KO cells belonging to C5 have not reached the full potential of HEM. To better understand this, we performed gene set enrichment analysis (GSEA) for C5 of Ldb1 KO vs. Ctrl. We detect several negatively enriched gene ontology terms and biological processes (GOBP) in Ldb1KO cells vs. Ctrl, that are crucial for HEM and early blood development (REV. Fig. 14 A). Those include positive regulation of BMP4 signalling, cell fate commitment and primitive erythroid lineage. The differences in gene expression of Ldb1 KO cells can ultimately reduce the full HEM potential of C5 and ultimately lead to a developmental block in Ldb1 KO cells. Concerning **Zbtb7b KO cells** (as shown in Fig. 4 d,e,) they have a **marked increase of cells belonging to C5** compared to Ctrl (**21.11% in Zbtb7b2 KO vs 5.3% in Ctrl**). This data is in agreement with our FACS quantifications of F_SP HEM, where Zbtb7b KO cells have a higher F_SP HEM frequency than Ctrl cells (as shown in Fig.3 a). For C6 (as shown in Figure 4 d,e), both genotypes reach similar levels (31.55% in Zbtb7b KO vs 35.63% in Ctrl). Furthermore, Zbtb7b KO cells show an accumulation of cells belonging to C4 (pre-HEM) (6.53% in Zbtb7b KO vs. 2.09% in Ctrl) (Supp.Fig.4b). Interestingly, the GSEA of C4 of Zbtb7b KO vs. WT revealed several GOBPs crucial for HEM differentiation and subsequent early blood development, which is in agreement with our findings that Zbtb7b acts as a repressor during HEM differentiation (REV.Fig.14 B).

REV Fig. 14

A Cluster 5 Ldb1 KO. vs Ctrl

Term	NES	p-adjust
positive regulation of BMP signaling pathway	-2.362190897	0.013400613
cell fate commitment	-2.303673333	0.012678423
primitive erythroid lineage	-1.440556139	0.033012023

B Cluster 4 Zbtb7b KO. Vs Ctrl

Term	NES	p-adjust
endothelial cells	2.893000806	1.00E-10
embryonic hemopoiesis	2.340795158	0.012572598
angiogenesis	2.328359459	0.021721066
hemopoiesis	2.318824694	0.021721066
vasculogenesis	2.282360567	0.021721066

Q16: 15. lines 356-357: "Zbtb7b-KO cells, which are largely composed of the same clusters as the Ctrl, formed the most advanced C6 (HEM 2) via a different trajectory" How was this determined? Has pseudotime analysis been performed?

Author response: Indeed, as outlined in the manuscript, pseudotime analysis was performed. For better clarity, we have expanded the methods section as follows:

Lines 770-776:

For pseudo-time analysis, the Seurat object was exported to a cds object using the SeuratWrap- pers package. Pseudotime trajectory analysis was conducted using Monocle3⁶⁴.

The trajectory graph was constructed on the previously established UMAP embeddings from Seurat via the 'learn_graph' function, using default options but 'use_partition' being set to false. Cluster 3 cells were designated as the root population based on their gene expression profile indicating a more immature state. Cells were ordered along this trajectory using the 'order_cells' function to establish pseudotime coordinates.

Q17. 16. lines 355-356: "For all analysed conditions, Zbtb7b KO mapped with the highest frequency on the in vivo trajectory (21.1% vs. 20.4% for Ctrl)" Can the authors explain what this statement means exactly? Do they consider this a significant difference?

Author response: We thank the reviewer for the comment regarding Supplementary Figure 4d. In this figure—similar to main Figure 1g—we mapped scRNA-seq data of FLK1⁺ cells isolated from day 5 embryoid bodies (EBs) of various knockout (KO) conditions onto a reference mouse gastrulation atlas (PMID: 30787436), which includes in vivo cells from embryonic day 6.5 to 8.5 (E6.5–E8.5). We subsequently performed a data integration onto a force-directed graph of the haemato-endothelial landscape established in the same study ((PMID: 30787436). For more technical details we refer to our response to minor point 2 below.

We interpret higher mapping frequencies onto the in vivo atlas as indicative of greater transcriptional similarity to in vivo embryonic mesodermal populations. The observed difference in mapping frequencies—21.1% for Zbtb7b KO versus 20.4% for control—is descriptive only and was not statistically assessed in this context. At present, we do not claim biological significance, and additional replicates would be required to determine whether this trend is consistent.

Notably, among the conditions tested, the Smad1 KO sample displayed the lowest mapping frequency at 16.5%, suggesting reduced similarity to in vivo counterparts under this condition. We include this observation as an illustrative comparison within the dataset and to highlight potential functional consequences of Smad1 deletion.

Q18: 17. I generally disagree with the authors' conclusions regarding Smad1 KO. The fewer live cells in Smad1 KO (Figure 4c) could be indicative of a more generic role of Smad1 than the one proposed here, i.e. a more profound effect on all mesoderm. Proper quantification and statistical tests should be reported for each cell type if they want to make points such as "Smad1 KO cells were able to form both haematopoietic and endothelial cells albeit at a reduced frequency and number" (lines 264-265)

Author response: We thank the reviewer for raising the point regarding the differentiation capacity of Smad1 KO cells. In our manuscript we perform two different differentiation assays, in the first one we evaluate mesodermal commitment (during EB culture) and in the second we differentiate the cells towards endothelial and haematopoietic lineages (HB culture). Figure 4 c shows the total number of FLK1⁺ cells after 2.5 day in HB culture. We detect a reduced number of live cells in the Smad1 KO compared to the control cultures. This difference is a trend and does not reach significance. At the moment, we did not evaluate the differentiation capacity of Smad1 KO cells in any additional differentiation system and we therefore cannot comment on any broader effect on Smad1 in other models, but rather describe the phenotypes detected in our assays. In REV Fig. 15 we quantify the frequency and total cell number of CD41⁺ cells (all haematopoietic lineages) (REV Fig. 15, A, B) and VE-cad⁺ cells (endothelial) (REV Fig. 15 C, D). Smad 1 KO cells show a reduced frequency of both cell types, reaching significance for CD41⁺ cells. These results are in agreement with the Fig.4 a-c shown in the manuscript. We therefore respectfully disagree with the reviewer and think our statement that 'Smad1 KO cells form both haematopoietic and endothelial cells albeit at a reduced frequency and number' is correct.

We additionally want to refer to Q.7A of the same reviewer, where we analyse Smad1 KO cells in EB cultures. Here we scored the total number of FLK1⁺ cells (mesodermal committed cells) (REV Fig. 10e, Supp. Fig.4e) and we detected a trend of reduced FLK1⁺ cells in Smad1 KO cells compared to Ctrl cell lines without reaching significance. We therefore cannot conclude that Smad1 has a general effect on mesodermal commitment in this assay.

REV Fig. 15

Q19: 18. Line 276, "Heterogeneity of mesodermal cells is regulated by core TF network". I am not sure what the authors mean with this. They are not really assessing any kind of network, but just the effect of independent mutations. The impact of those mutations on gene expression or accessibility of other TFs is not analyzed.

Author response: We thank the reviewer for this valuable comment and apologise if our original phrasing was misleading. We acknowledge that we did not experimentally assess how the newly identified TFs functionally interact, and we therefore agree that referring to a "TF network" in the manuscript may have been premature. As suggested, we have removed any references to a TF network. Furthermore, to explicitly highlight this limitation of our study we have now included the following sentence in our discussion:

Lines 493- 495:

"Importantly, despite our efforts to analyse the gene expression perturbations resulting from loss-of-function of all the validated candidates, mechanistic understanding of the gene function and **of the interplay between the identified TFs is needed in the future**".

Minor comments

1. line 98- typo: Ldb1 is the correct gene name.

Author response: The typo was corrected.

2. lines 150-152: "Finally, to further underline the relevance of our ESC model we mapped the in vitro-generated cells from both EB culture and haemangioblast culture onto a mouse gastrulation atlas of cells isolated from E6.5–E8.5 embryos". What kind of data were mapped here?

Author response: We mapped scRNA-seq data from our in vitro-generated cells (from both EB and HB cultures) onto a mouse gastrulation atlas of cells isolated from E6.5–E8.5 embryos (PMID: 30787436). This integration was performed using the Seurat package as detailed in our methods section (SORT-seq) as follows:

Integration of selected haemato-endothelial cells from the publicly available mouse gastrulation atlas²⁸ with our in vitro scRNA-seq dataset was performed using Seurat. To complement this integration, the force-directed graph of the haemato-endothelial landscape was reconstructed by incorporating supplementary data from the atlas publication. Within the integrated dataset, in vitro population labels were assigned to in vivo cells from the atlas based on the k=5 nearest neighbours.

We have revised the text as follows:

lines 173-175:

Lastly, to further underline the relevance of our ESC model, we integrated the transcriptional profiles of our in vitro-generated cells from both the EB and HB cultures with the mouse gastrulation atlas²⁸, which includes cells isolated from E6.5–E8.5 embryos.

RESPONSE TO REVIEWER COMMENTS_2nd round

We thank the reviewers for their constructive comments and valuable suggestions on our previous manuscript version and are pleased that R1 and R2 are fully satisfied with our first round of revision. We have carefully addressed the remaining points raised by R3 in the following point-by-point response. To support our revision, we provide additional analysis of our data and clarified conclusions throughout the revised manuscript.

To address these revisions, we have added updated figures/ figure panels to the revised manuscript:

- New panel in Figure 1: e
- New Supplementary Figure 3

Additionally, we updated Figure 1h as we detected a mistake in the colour scale.

The new additions and responses to the reviewers' comments can be found in the point-by-point response below.

Reviewer #1 (Remarks to the Author):

The authors have made an earnest and effective attempt to address the critiques from the prior review and have added data that addresses many of the points raised. The addition of some discussion of the limitations of the study has been incorporated. I agree with the revisions based on the very legitimate criticism raised by reviewer #3 of the lack of true analysis of a "transcription factor network".

Author response: We thank the reviewer for their comment and acknowledging our contribution.

Reviewer #2 (Remarks to the Author):

All my comments are adequately addressed.

Author response: We thank the reviewer for their comment and acknowledging our contribution.

Reviewer #3 (Remarks to the Author):

I have reviewed the revised version of the manuscript titled "Targeted CRISPR-Cas9 screening identifies core transcription factors controlling murine haemato-endothelial fate commitment" by Teske et al.

I appreciate the efforts made by the authors to respond to reviewer comments. Although a number of issues raised by this and other reviewers have been addressed to a certain extent by including additional data, information, or discussion, several concerns remain.

The main experimental addition to the revised manuscript was a new "long-term" (9.5 days) hematopoietic differentiation assay, showing that F_{SP} HEM cells have significantly higher blood-forming potential compared to DP priMes, as expected. They quantified CD45⁺ and CD71⁺ output, confirming the already known biological significance of the PDGFR α downregulation as a developmental transition. Note that the materials and methods section titled "Long-term haematopoietic assay in liquid media" appears twice (lines 622-634 and 685-692), with different content.

No in vivo validation was performed for any of the factors emerging from their screen. Although the authors have now acknowledged this limitation in the Discussion, I believe that they should have at least tried. As is, the validity and significance of the findings remains limited to an in vitro murine system.

I found the comment about phenotype rescue from one of the other Reviewers particularly relevant. In the absence of in vivo validation, the claim that the TFs identified are important regulators of mesodermal commitment would be indeed strengthened if they at least showed some sort of functional rescue. The authors tried to address the comment and attempted two lines of rescue: (i) generation of double knockouts (e.g., Ldb1/Zbtb7b), showing only a partial phenotypic – but not functional – rescue (no blood or even endothelial cells generated) and (ii) lentiviral ETV2 overexpression in various KO lines. Again, the phenotype observed was only an increase of VE-Cad⁺ cells and only in some knockouts. These results are very preliminary and were not included in the revised article. Thus, I feel that at present, the functional relevance of the identified factors as regulators of HEM specification is not completely supported by the data and this important concern has not been conclusively addressed.

Major comments and outstanding issues:

1. I appreciate that the authors have recognized that, as pointed out in my previous review, some methodology was completely missing from the manuscript (ie. Spectral flow cytometry); they have now added information in their revised manuscript. They have now added information regarding how they have defined their populations after HB culture by spectral flow (Revised Fig 1f) (lines 148-155). I find these definitions very arbitrary and questionable. For instance, why do "EMPs" do not express CD16/32 (see <https://pubmed.ncbi.nlm.nih.gov/26095363/>) and are largely positive for CD43 (stem/progenitor marker) and CD71? This is a phenotype more consistent with erythroid progenitor activity, if anything. It is also puzzling that cells annotated as primitive erythrocytes and macrophages express CD41 as stated in the text, though this does not seem to correspond to what shown in the bubble plot (only a fraction of these cells appears to be CD41+). It is also strange that primitive macrophages appear largely positive for CD43, a marker associated with late progenitor/preHSC activity in mouse (<https://pubmed.ncbi.nlm.nih.gov/25241746/>). In general, the annotated populations seem very heterogeneous.

Author response: We thank the reviewer for the helpful comment and the opportunity to clarify concerns regarding our spectral flow analysis of HB culture at day 2.5 (Fig. 1f, g of revised manuscript). The primary objective of this analysis was to compare the differentiation capacity of F_SP HEM with that of DP priMes. Regardless of the nomenclatures of cell clusters, we found an increased potential of F_SP HEM to form haematopoietic and endothelial lineages (mainly defined by CD41, CD71, c-kit and VE-cad expression) in comparison to DP priMes **in agreement to our conventional FACS analysis** of Figure 1d, e.

We acknowledge the reviewer's concern that the nomenclature of clusters may be misleading if interpreted as representing fully differentiated haematopoietic cells. **Haematopoietic populations differentiating in HB cultures at day 2.5 are not fully mature but represent progenitor states** (e.g. PMID: 18339678; PMID: 36217016). Therefore, to avoid misinterpretation, we will revise our nomenclature as follows:

Erythroid-myeloid progenitors (EMP) → **definitive haematopoietic progenitor cells** (dHPC)
Primitive erythrocytes → **primitive erythrocyte progenitors** (pEryP)
Primitive macrophages → **macrophage progenitors** (MacP)

While we appreciate the reviewer's comment, **we respectfully disagree with the criticism concerning marker expression observed in the differentiated haematopoietic cells**. As our data concerns a developmental process the expression of specific markers depends on the maturation state of the cells.

The erythroid-myeloid **progenitor** (EMP) cluster (now termed dHPC in the revised manuscript) was mainly defined by CD41^{hi}/c-kit⁺/CD43⁺/CD71⁺ expression. In contrast to the publication cited by the reviewer, we did not detect CD16/32 expression on our EMPs. The mentioned publication employs a different ESC-based haematopoietic differentiation system (prolonged EB culture) where EMPs express CD16/32. **CD16/32 expression depends on the maturation status of EMP that are initially CD16/32 negative** (PMID: 26418893). It was shown that CD41⁺c-kit⁺ CD16/32⁻ haematopoietic progenitor cells mature into CD41⁺c-kit⁺CD16/32⁺ EMPs (PMID: 26418893). Therefore, the differences between our study and the referred publication could simply stem from different EMP maturation status/ time point of analysis. CD41 and c-kit, the other two major markers, are expressed in EMPs in our model and in the publication referenced by the reviewer. Additionally, our EMP classified cells are positive for CD43, which is a **marker found on nearly all haematopoietic colony-forming cells in the YS, AGM and foetal liver tissues as early as E9.5** (PMID: 24749071). CD71 expression is furthermore a phenotype commonly found in other publications that employ similar in vitro ESC-derived haematopoietic differentiation models (e.g. PMID: 31015568) and we agree with the reviewer that CD71 expression points to an erythroid bias. In the revised manuscript, we rename the EMP cluster as dHPC. This terminology has been used in previous studies to describe CD41⁺c-kit⁺ haematopoietic progenitors (e.g., PMID: 12393529; PMID: 31015568) and, importantly, also encompasses an 'immature' EMP phenotype."

CD41 is the earliest haematopoietic marker which is transiently expressed and downregulated after haematopoietic commitment and upon HPC maturation (PMID: 11934866; PMID: 12393529; PMID: 12900455). We sincerely apologise for the mistake we made in the text which was indeed misleading, and

we corrected it to: CD41^{+/−}, indicating that primitive erythrocyte progenitors, macrophage progenitors and to a lesser extent dHPC are composed of both CD41⁺ and CD41[−] cell fractions.

Regarding the macrophage cluster expression profile, as stated above, CD43 expression is found on nearly all haematopoietic colony-forming cells in the YS, AGM and foetal liver tissues as early as E 9.5 (PMID: 24749071). Macrophage progenitors were further identified through expression of key markers including: CD11b/ CD11c, CX3CR1, and CD45. The reviewer refers to a publication concerning HSC development in the AGM region and not yolk sac haematopoiesis (HSC independent), which is what we aim to recapitulate in our model system.

Finally, **we defined 9 major clusters separating endothelial, haematopoietic and VSM-enriched lineages**, without further subdividing each major cellular cluster following biological reasoning. Our spectral flow analysis aims to dissect the differentiation capacity of distinct mesodermal subpopulations to form the mentioned main lineages. We do not find it useful to further subdivide those clusters. Our rationale for employing a **data-driven strategy** is to capture the full spectrum of marker expression simultaneously and to **apply dimensionality reduction methods that unbiasedly resolve cellular clusters in a high-dimensional space**. This approach avoids the limitations inherent in investigator-driven gating of individual markers and provides a more comprehensive and reproducible framework. To directly address the reviewer's concern, we include below representative plots illustrating a conventional manual gating strategy (including heterogenous marker expression for some clusters) (**REV2_Fig1**). Should the editors deem it useful, we will gladly provide these as a supplementary figure. The depicted manual gating strategy is **solely to visualise** our cells and was **not** used to define the clusters.

REV2_Fig.1

2. Relative to my previous comment no.3, I feel that my concern (quantification of data in Fig1) has not been properly addressed. Moreover, no quantification has been reported for the endothelial potential, thus not supporting the statement "The HB culture experiments confirmed that DP priMes indeed had less capacity to form haematopoietic and endothelial lineages compared to F_SP HEM, which generate a higher frequency of CD41+ and VE-cad+ cells and fewer VSM-enriched cells" (lines 140-143)

Author response: We thank the reviewer for the comment that concerns Figure 1d/f addressing the differentiation potential of mesodermal populations toward haematopoietic (CD41⁺) and endothelial (VE-cad⁺) lineages.

We included a representative FACS analysis in our original manuscript as Fig.1d (REV2_Fig.2a) but acknowledge that we did not provide a quantification of the frequency of VE-cadherin single-positive (SP) cells, but only CD41 SP and CD41⁺/VE-cad⁺ double-positive (DP) (REV2 Fig.1b). We have now added the missing quantification of VE-cad SP cells with a new panel in Figure 1e of the revised manuscript (REV2_Fig.1b, right panel). In addition to the frequencies, we also provide here the corresponding cell numbers for each population (REV2_Fig. 2c). Each HB culture was started with the exact same number of mesodermal cells (either DN, P_SP, DP priMES, or F_SP HEM) and HB cultures started from **F_SP HEM** demonstrated the **highest capacity to generate both CD41⁺ (haematopoietic) and VE-cad⁺ (endothelial) cells—both in terms of frequency and absolute cell numbers—when compared with the other mesodermal populations** (REV2 Fig. 1a-c), therefore **justifying our original statement regarding the superior potential of F_SP HEM in forming haematopoietic and endothelial lineages.**

REV2 Fig.2

3. Relative to my previous comment no.4. While not central to the main message of the study, I disagree with the statement included in the revised article that "there is no specific marker for VSM cells suitable for flow cytometry". What about aSMA, NG2, CD146, PDGFRb..? They might not be the more specific markers, but the inclusion of at least some of these markers would definitely help demonstrating the non-hemato-endothelial identity of the cells that they call vSMCs. Also, they say their panel is designed for hemato-endothelial cells, and therefore the identification of vSMC is based on the exclusion of lineage markers, but this is in contrast with the expression of hemato-endothelial markers on those cells (CD43, CD44, CD16/32, CD14...).

Author response: We agree with the reviewer that the most specific marker for VSM cells is alpha smooth muscle actin (aSMA, encoded by *Acta2*). However, analysis of aSMA by flow cytometry requires intracellular staining, which we did not perform in this study, as permeabilization can compromise the signal of other surface markers. We acknowledge that our previous wording may have been misleading. We have now revised the text to state: "there is no specific **surface** marker for VSM cells suitable for flow cytometry" (lines 138-139 in the revised manuscript). As detailed in our earlier response (R2, point 2), and consistent with

other publications (e.g. PMID: 31996681), we classify cells that lack both CD41 and VE-cad expression as belonging to the VSM-enriched cluster and did not include additional markers like NG2, CD146 and PDGFR β . That CD41⁻VE-cad⁻ cells form the VSM-enriched cluster is **further validated by our single-cell RNA-seq analysis** (employing SORT-seq). During SORT-seq, single-cells were FACS sorted based on FLK1, CD41, VE-cad, and c-kit expression from EB and HB stage. The gating strategy of the single-cell FACS sort is depicted in REV2_Fig. 3a below. **CD41⁻VE-cad⁻ cells form a separate cluster** (REV2 Fig. 3b) that is characterised by expression of **key VSM-associated genes**, including *Acta2* (aSMA), *Pdgfrb*, and *Tagln*, while **lacking endothelial and haematopoietic markers** (e.g. *Tek/Tie2*, *Pecam/CD31*, *Sox17* (expressed on VE-cad⁺ cells) and *Tal1*, *Gfi1b*, *Runx1* (expressed on CD41⁺ cells)) (REV2_Fig3c). Therefore, the **CD41⁻VE-cad⁻ cell cluster** was named VSM-enriched. To further alleviate concerns regarding our model we now provide a new Supp.Fig. 3 in our revised manuscript extending our data on the scRNA-seq characterisation (including the gating strategy of the single-cell sort, cell clustering and marker gene expression profiles within cell populations). These data complement Fig. 1c, h and Supp. Fig. 1c, which were already included in the previous version and concern the same scRNA-seq dataset.

We included the following text in our revised manuscript (lines 174-184):

"Lastly, to further underline the relevance of our ESC model, we performed a scRNA-seq characterisation of both stages of our model employing SORT-seq. By relying of the lineage defining markers FLK1, CD41, VE-cad and c-kit, we isolated single cells corresponding to 7 distinct populations from the EB and HB stage (Supp. Fig 3a). Our analysis confirmed the generation of distinct lineages characterised by appropriate signature gene expression patterns, including pHPCs (primarily primitive erythrocyte progenitors), dHPCs, VSM-enriched cells and endothelial lineage (Supp.3b-d). Additionally, we integrated the transcriptional profiles of our in vitro-generated cells with a mouse gastrulation atlas²⁸, which includes cells isolated from E6.5–E8.5 embryos. A high proportion (39.8%) of the in vitro-generated cells could be directly projected onto the in vivo haemato-endothelial trajectory graph, evidencing the similarities between in vitro- and in vivo-generated cells (Fig. 1h)."

REV2_Fig3

4. Relative to my previous comment no. 19. The manuscript still does not contain any TF network analysis. Although the authors agreed with this and removed most references to "TF networks," there are still instances in the manuscript where reference to transcriptional networks appear, e.g.: end of introduction (line 99); Title of section referring to Figure 4; Figure 5 title.

Author response: We thank the reviewer for this useful comment and removed the mentioned references to a transcriptional network in our revised manuscript.

5. Relative to my previous comment 17. "For all analysed conditions, *Zbtb7b* KO mapped with the highest frequency on the in vivo trajectory (21.1% vs. 20.4% for Ctrl) indicating their enhanced capacity to commit towards the haemato-endothelial lineage (Supp. Fig. 4d)." As the sentence is descriptive and not biologically significant, and that difference it's really not relevant from an n=1 as confirmed by the authors in their response, the claim and this sentence should be removed.

Author response: We removed the sentence in our revised manuscript.

Other comments:

Line 177: reference to fig. 1f appears to be wrong.

Author response: We apologise for this mistake. The correct figure panel is Fig.1h. This has been corrected in the revised manuscript.

Line 202: reference to supp. Table 1 is wrong. Suppl. Table 2 appears to be missing.

Author response: We apologise for this mistake. The correct table is Supp. Table 2. This has been corrected in the revised manuscript, and Supp. Table 2 has been uploaded.